# MoSE: Decoupled Tuning for Forgetting-Resilient Multi-task Fine-tuning of LLMs

## Abstract

Integrating Low-rank Adaptation (LoRA) and Mixture-of-Expert (MoE) is the mainstream for applying LLMs in multi-task scenarios. Existing works assume that different experts can share common knowledge and hold the specific information dynamically. Thus, they employ router to select appropriate experts for different tasks. Despite the achieved progress, most of them tune LoRA modules indiscriminately, which will cause the learned information in LoRA from previous task to be overwritten by the fine-tuning of subsequent tasks. Therefore, existing works still face the problem of cataclysmic forgetting of both common and specific information in LoRA. To tackle this problem, in this paper, we propose a novel Mixture of Shared and Exclusive Experts framework (*MoSE*) for better multi-task fine-tuning of LLMs. Different from most existing works, we first separate the LoRA experts into routing experts for task-specific information and shared experts for common knowledge. For routing experts, we develop a feature-wise module to select the most appropriate experts and tuning their parameters entirely. For shared experts, we aim to maintain as much common knowledge as possible. Thus, we design a novel Top-$k$-selection tuning strategy to selectively fine-tune certain parameters of shared experts. Then, we adopt a balanced data sampling and expert assignment strategies to mitigate task imbalance and ensure fair expert utilization. Finally, we conduct extensive experiments over diverse multi-task scenarios to demonstrate the effectiveness of *MoSE*. Moreover, *MoSE* exhibits strong continual learning ability, effectively adapting to new tasks while retaining prior knowledge (average 3.3% and 7.4% improvement compared with advanced baselines in sequential continual learning).

## 1 Introduction

The success of Deepseek-V3 (Liu et al., 2024a), and Qwen3 (Yang et al., 2025a) has proven the potential of Mixture-of-Expert (MoE) structure, inspiring the integration of Low-Rank Adapter (LoRA) and MoE for general multi-task solver tuning. Various methods have been proposed to improve the generalization capability of LLMs across multi-task scenarios, such as MixLoRA (Li et al., 2024), MultiLoRA (Wang et al., 2023a), MOELoRA (Liu et al., 2024c), and Lorahub (Huang et al., 2024). They all have achieved impressive performance in multi-task scenarios.

By comparing these works, the basic motivation is to use different experts to memorize knowledge from different tasks. Then, a task router is employed to activate appropriate experts for target tasks, reducing the computational cost and inference latency. Moreover, these works claimed that the expert overlap among different tuning steps can also be treated as sharing common task knowledge. For example, HMoRA (Liao et al., 2025) uses sparse routing to implicitly share experts across tasks with similar demands. In contrast, ViMoE (Han et al., 2024) augmented sparse experts with an additional shared expert, leveraging routing strategies tailored to vision tasks to stabilize training and separate global from specialized representations. Both approaches achieve promising performance in their respective domains.

In practice, we argue that common knowledge should be preserved as much as possible, requiring fewer updates to avoid overwriting, while task-specific information needs to be acquired rapidly through accurate expert selection. However, most existing works fail to satisfy this requirement. For one thing, the router does not memorize the roles of selected experts. Therefore, the expert with

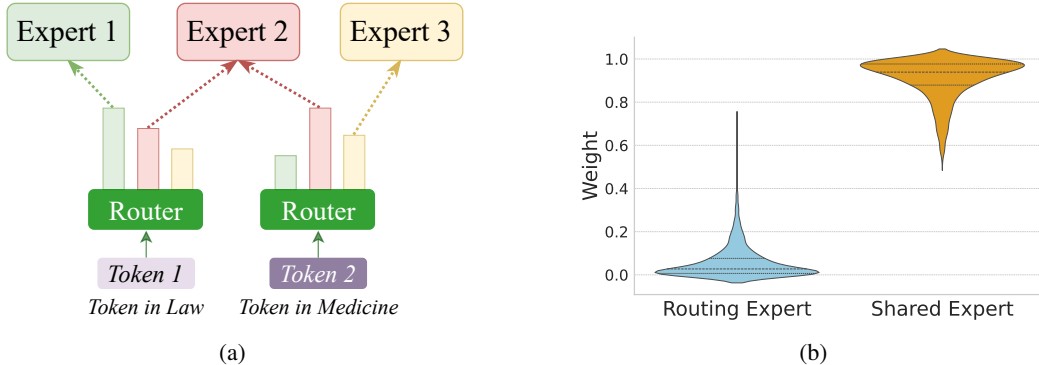

Figure 1: (a) Router selects experts for tasks while expert roles are not distinguished. (b) Explicit expert roles where shared expert dominates due to excessively large weights.

common knowledge in task A might be selected to learn task-specific information when facing task B, as shown in Figure 1(a). This situation do not only cause the catastrophic forgetting of knowledge from previous tasks, but also lead to high tuning cost due to the ineffective common knowledge sharing across different tasks. For another, even some works use shared experts to memorize common knowledge, they still use the same strategy to tune the entire shared experts with routing experts at each step, leading to larger weight in inference phrase and limited impact of other experts. Taking Figure 1(b) as an example, in the multi-task GLUE setting the shared expert tends to receive larger weights during inference, thereby dominating the prediction process and suppressing contributions from the routing experts. These phenomena raise an important question: "**How to achieve effective sharing of common knowledge while retaining task-specific information when fine-tuning LLMs in multi-task scenarios?** "

To tackle the above problem, in this paper, we propose a novel Mixture of Shared and Exclusive Experts framework (*MoSE*) for better multi-task fine-tuning of LLMs. Following the advanced MoE framework, *MoSE* separates LoRA experts into shared and routing experts to capture common knowledge and task-specific information. For routing experts, we design a feature-wise gating mechanism to select suitable experts for task-specific learning. For shared experts, considering the target of memorizing and sharing common knowledge, we develop a novel Top-$k$-selection strategy to selectively updates parameters with consistent gradients. Thus, the catastrophic forgetting and tuning frequency problems can be well mitigated. Moreover, to address the imbalance in routing expert selection, we adopt a balanced expert assignment strategy to stabilize the tuning process. Finally, we conduct extensive experiments over multiple LLMs across various multi-task scenarios. Experimental results demonstrate that our proposed *MoSE* can maintain as much common knowledge as possible and surpass other advanced baselines by a large margin. Specifically, *MoSE* achieves average improvements of 3.3% and 7.4% over MixLoRA and MultiLoRA in sequential continual learning, demonstrating stronger knowledge sharing and greater resistance to forgetting.

## 2 RELATED WORK

### 2.1 PARAMETER-EFFICIENT FINE-TUNING

Parameter-Efficient Fine-Tuning (PEFT) adapts large language models (LLMs) to downstream tasks by updating only a small subset of parameters. Representative approaches include Prompt Tuning (Lester et al., 2021), Prefix Tuning (Li & Liang, 2021), Adapters (Houlsby et al., 2019), and LoRA (Hu et al., 2022). Prompt- and Prefix-based methods inject trainable vectors into the input, whereas Adapters and LoRA insert lightweight modules into the network for internal adaptation. Among them, LoRA has become particularly popular for its efficiency and scalability, as it decomposes weight updates into two trainable low-rank matrices. Building on this idea, several extensions have been proposed: AdaLoRA (Zhang et al., 2023a) dynamically allocates rank budgets; LoRA+ (Hayou et al., 2024) applies separate learning rates; MoSLoRA (Wu et al., 2024) introduces a learnable mixer; and RoSE-LoRA (Wang et al., 2024a) enforces sparsity to update only critical parameters. Although these methods are effective in single-task settings, they rely on a fixed shared subspace,

which restricts flexibility and degrades performance in multi-task scenarios (Wang et al., 2023a). This limitation underscores the need for more adaptive and disentangled architectures that can better capture task-specific variations while still leveraging shared knowledge.

## 2.2 MULTI TASK LEARNING

Multi-Task Learning (MTL) trains a unified model across tasks to improve generalization and data efficiency (Zhang et al., 2023b), but traditional approaches often struggle with task heterogeneity and gradient conflict (Liu et al., 2019). Parameter-Efficient Fine-Tuning (PEFT) methods have recently gained traction in MTL for their scalability and low overhead. Among them, LoRA is widely adopted, yet its reliance on a shared subspace can cause interference when tasks diverge (Yang et al., 2024; Zhao et al., 2025). To alleviate this, recent work integrates LoRA with Mixture-of-Experts (MoE): MultiLoRA (Wang et al., 2023a) employs parallel LoRA modules with routing, MoELoRA (Liu et al., 2024c) applies task-specific gating, AdaMoE (Zeng et al., 2024) enables token-adaptive routing with null experts, and MoLA (Gao et al., 2024) prioritizes higher-layer expert allocation. To further address these limitations, our approach introduces a structured design with a shared expert for common knowledge and routing experts for task-specific features.

## 2.3 CONTINUAL LEARNING

Continual Learning (CL) aims to train models sequentially on multiple tasks without catastrophic forgetting—the loss of previously acquired knowledge (Wang et al., 2024b). Classical approaches include regularization-based methods such as Elastic Weight Consolidation (Kirkpatrick et al., 2017), which protect important parameters, and replay-based strategies that rehearse prior data (Rebuffi et al., 2017). While PEFT has gained popularity for efficient adaptation, most methods remain single-task oriented and lack mechanisms for knowledge preservation, often resulting in severe forgetting. To address catastrophic forgetting, several PEFT-based continual learning methods have been proposed. I-LoRA (Li et al., 2025) employs a dual-memory design, where a fast branch quickly adapts to new tasks while a slow branch preserves stable representations. GainLoRA (Liang & Li, 2025) introduces gated feature integration to fuse old and new information, and NTK-CL (Liu et al., 2024b) leverages orthogonal EMA-smoothed gradients to minimize interference with past tasks. Although *MoSE* is not specifically designed for continual learning, its architectural characteristics share conceptual similarities with methods such as I-LoRA and EWC. Therefore, we posit that *MoSE* can effectively retain general knowledge, reduce cross-task interference, and consequently enhance the model's resistance to forgetting.

## 3 TECHNICAL DETAILS OF *MoSE*

### 3.1 LOW-RANK ADAPTION

LoRA (Hu et al., 2022) enables parameter-efficient fine-tuning by injecting low-rank adapters into pre-trained weights. Instead of updating the full matrix $W_0 \in \mathbb{R}^{d_{\text{out}} \times d_{\text{in}}}$, it adds a low-rank update during the forward pass as follows:

$$h = W_0 x + \alpha B A x, \tag{1}$$

where $A \in \mathbb{R}^{r \times d_{\text{in}}}$, $B \in \mathbb{R}^{d_{\text{out}} \times r}$, and $r \ll \min(d_{\text{in}}, d_{\text{out}})$. This significantly reduces the number of trainable parameters while preserving model expressiveness.

### 3.2 *MoSE* FRAMEWORK

Figure 2 illustrates the overall structure of our proposed *MoSE*, including a structurally explicit shared expert and multiple routing experts. To achieve the efficient common knowledge sharing and task-specific information keeping, we develop different learning strategies for different experts. Next, we will introduce each of them in detail.

### 3.2.1 ROUTING EXPERT LEARNING

Similar to previous work, we intend that routing experts should be task-sensitive. Thus, we focus on using task information to dynamically select appropriate experts with a task router. Specifically, we

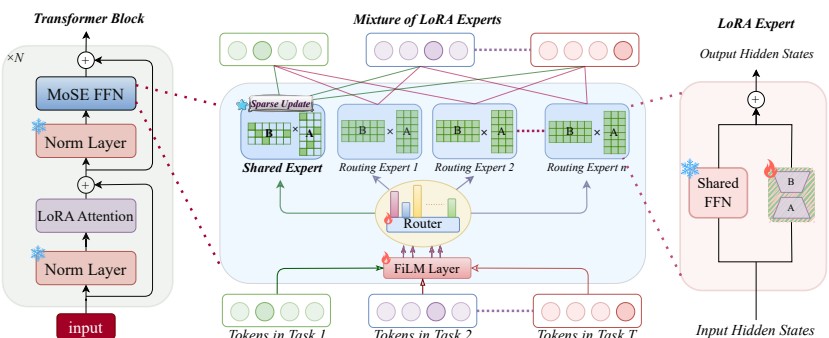

Figure 2: The architecture of our proposed *MoSE* framework.

first construct task embedding $e_t$ by averaging the final-layer hidden states of 20 randomly selected training samples. This operation avoids task-specific optimization and offers a lightweight, transferable alternative to approaches like MOELoRA (Liu et al., 2024c), which is in favor of improving the generalization capability of *MoSE*.

After that, we employ Feature-wise Linear Modulation (FiLM) to condition input representations on task embedding $e_t$, so that the router can be task-aware, which can be formulated as follows:

$$x^{\text{mod}} = x \odot (1 + \gamma(e_t)) + \beta(e_t), \tag{2}$$

where $\gamma(\cdot)$ and $\beta(\cdot)$ are learned affine functions producing scale and shift parameters, and $\odot$ denotes element-wise multiplication. One step further, to reduce the overhead of a fully parameterized FiLM layer ($\mathcal{O}(d_t \cdot d_h)$ parameters), we adopt a **low-rank FiLM** structure by decomposing the transformation into two linear projections:

$$[\gamma(e_t), \beta(e_t)] = W_{\text{up}} \cdot (W_{\text{down}} \cdot e_t), \tag{3}$$

where $W_{\text{down}} \in \mathbb{R}^{r \times d_t}$ and $W_{\text{up}} \in \mathbb{R}^{2d_h \times r}$ with $r \ll d_h$. This reduces parameters from $2d_t d_h$ to $2r(d_t + d_h)$, significantly improving efficiency while preserving expressiveness. Moreover, FiLM is a lightweight, plug-and-play module for task-aware conditioning, readily adaptable to diverse routing strategies and backbone architectures without structural modifications.

After obtaining the task-aware representation $x^{\text{mod}}$, we apply *Top-K Routing* to compute expert scores and select appropriate experts. Specifically, we first compute routing logits $\mathbf{z} = W_r x^{\text{mod}} + b_r \in \mathbb{R}^{K+1}$, where $W_r \in \mathbb{R}^{(K+1) \times d}$ and $b_r \in \mathbb{R}^{K+1}$ are learnable parameters. Among the $K+1$ scores, we retain the score $z_0$ for the shared expert, and select the top-$k$ scores from the remaining $K$ routing experts. The final routing weights are obtained via a softmax over these $k+1$ scores:

$$\mathbf{w} = [w_0, w_1, ..., w_k] = \text{softmax}\left(\text{concat}\left(z_0, \text{TopK}(\mathbf{z}_{1:K}, k)\right)\right), \tag{4}$$

where $\mathbf{z}_{1:K}$ denotes the logits for the routing experts. The resulting $\mathbf{w} \in \mathbb{R}^{k+1}$ provides normalized routing weights over the shared expert and the selected routing experts. The final output of the *MoSE* module is computed as a weighted combination of the corresponding expert outputs:

$$\text{MoSE}(x) = w_0 \cdot \mathcal{E}_s(x) + \sum_{i=1}^{k} w_i \cdot \mathcal{E}_{j_i}(x), \tag{5}$$

where $\mathcal{E}_s(\cdot)$ denotes the shared expert and $\{\mathcal{E}_{j_1}, \ldots, \mathcal{E}_{j_k}\}$ are the top-$k$ routing experts. For routing experts, we use the same updating strategies with existing work (Zeng et al., 2024; Li et al., 2024). For shared experts, since our target is to memorize and share as much common knowledge as possible, we design a novel *Top-k selection* updating strategy, as detailed in the next section.

### 3.2.2 SHARED EXPERT LEARNING

As mentioned in Section 1, the shared expert is designed to facilitate efficient knowledge transfer and parameter reuse. However, differences in objectives, input distributions, and semantics often lead to gradient conflicts across tasks, producing compromises that fail to capture cross-task invariances (Zhang et al., 2023b). Therefore, the key challenge for shared experts is managing conflicting gradients from diverse tasks. A natural solution is to identify more consistent gradient directions

across tasks, thereby reducing conflicts and enabling stable knowledge integration. Thus, the problem turns to "how to determine the updating parts in each learning process". In response, we design a simple but effective strategy: **Top-$k$ selection**, which relies on parameter importance estimation.

For *parameter importance estimation*, we aim for more reliable assessments. Rather than relying on raw gradient magnitudes, we adopt an Exponential Moving Average (EMA) of gradients as a first-moment estimator, capturing the long-term importance of each parameter:

$$m_i^{(t)} = \eta \cdot m_i^{(t-1)} + (1 - \eta) \cdot g_i^{(t)}, \tag{6}$$

where $g_i^{(t)}$ denotes the gradient of the $i^{th}$ parameter at step $t$, $\eta \in (0,1)$ is the hyper-parameter of EMA, and $m_i^{(t)}$ represent its momentum. Parameters with large momentum magnitudes are viewed as consistently influential across training steps and are therefore prioritized. Moreover, larger gradient values generally reflect directions endorsed by most samples within a batch, which helps reduce conflicts and stabilize updates. Based on this, *Top-k selection* selects at each step the $k_{\text{top}}$ parameters with the highest importance scores:

$$\mathcal{S}_{\text{update}} = \text{TopK} \left( \{|m_i^{(t)}|\}_{i=1}^d, \ k_{\text{top}} \right). \tag{7}$$

Here, $\mathcal{S}_{\text{update}}$ denotes the subset of trainable parameters of $\mathcal{E}_s$ in Eq. 5, which are selected for updating. The remaining parameters of $\mathcal{E}_s$ are frozen during each gradient step. Moreover, to enhance tuning stability, we adopt a gradual sparsity schedule: training starts with a dense warm-up where all parameters are updated, then gradually reduces the number of updated parameters $k_{\text{top}}$ via a cosine decay, thereby increasing sparsity over time. This smooth transition enables the model to shift from full updates to the target sparse regime.

### 3.2.3 BALANCED EXPERTS LOADING

Apart from expert design, routing–expert selection also plays a crucial role in determining final performance. To strengthen *MoSE*'s effectiveness, we further introduce strategies that regulate this process. Unbalanced expert utilization is a common issue in MoE-based architectures, especially under top-$k$ routing, where certain experts are disproportionately activated. To address this, we propose two complementary regularization strategies—*Balance Constraint* and *Orthogonality Constraint*—to promote both fair and efficient expert utilization.

For *Balanced Constrain*, we focus on the routing experts and apply the balanced strategy (Li et al., 2024) to encourage uniform token-to-expert assignment. Specifically, given a training batch with $T$ tokens and $N$ experts, let $F_i$ denote the fraction of tokens routed to expert $i$, and $P_i$ the average router probability assigned to expert $i$. The balanced constrain is computed as:

$$\mathcal{L}_{\text{aux}} = N \cdot \sum_{i=1}^N F_i \cdot P_i. \tag{8}$$

For *Orthogonality Constrain*, we argue that shared expert should learn complementary and diverse representations, compared with routing experts. Therefore, we adopt an orthogonality regularization, which penalizes similarity between down-projection matrices to enhance modular disentanglement. Let $A_i \in \mathbb{R}^{r \times d}$ denotes the down-projection matrix of the $i^{\text{th}}$ routing expert and $A_{\text{shared}}$ that of the shared expert. The orthogonality loss is defined as:

$$\mathcal{L}_{\text{orth}} = \sum_{i=1}^N \left\| A_i A_{\text{shared}}^\top \right\|_1, \tag{9}$$

where $N$ is the number of routing experts and $\|\cdot\|_1$ denotes the element-wise $\ell_1$ norm. A lower $\mathcal{L}_{\text{orth}}$ indicates reduced directional overlap, enabling the shared expert to learn more task-agnostic features while routing experts specialize in task-specific knowledge.

### 3.2.4 OPTIMIZATION

Based on the above, our overall objective is defined as a weighted sum of the primary task loss $\mathcal{L}_{\text{task}}$, the balanced constraint $\mathcal{L}_{\text{aux}}$, and the orthogonality constraint $\mathcal{L}_{\text{orth}}$, with hyperparameters $\lambda$ and $\beta$:

$$\mathcal{L}_{\text{total}} = \mathcal{L}_{\text{task}} + \lambda \cdot \mathcal{L}_{\text{aux}} + \beta \cdot \mathcal{L}_{\text{orth}}. \tag{10}$$

In summary, grounded in our decoupled design philosophy, we introduce a shared expert into the MoE-LoRA framework. To enable this shared expert to effectively capture cross-task general knowledge, we employ a gradient-based importance–driven sparse update mechanism. Meanwhile, to ensure that the routing experts learn task-specific information, we incorporate a task-aware routing strategy. The combination of shared and routing experts allows the model to maintain strong representational capacity while achieving robust stability and resistance to cross-task interference.

## 4 EXPERIMENT

In this section, we evaluate *MoSE* across various scenarios and answer the following questions:

RQ1: Is *MoSE* effective in multi-task scenarios?

RQ2: Is *MoSE* capable of mitigating catastrophic forgetting?

RQ3: Do shared and routing experts perform as expected?

### 4.1 EXPERIMENTAL SETUP

**Evaluation Benchmarks.** To comprehensively evaluate *MoSE*, we design two scenarios: 1) *Multi-task learning and transfer*: Models are fine-tuned and evaluated on the GLUE benchmark (Wang et al., 2018) and commonsense reasoning tasks. We further assess transferability by testing commonsense-tuned models on unseen QA datasets, including CSQA (Talmor et al., 2019), CSQA2 (Talmor et al., 2021), LogiQA (Liu et al., 2021), and QASC (Khot et al., 2020). 2) *Continual learning*: We measure both catastrophic forgetting and backward transfer across QA tasks (CSQA, CSQA2, LogiQA), as well as SciTail (Khot et al., 2018), RACE (Lai et al., 2017), FOMC (Shah et al., 2023), and MedMCQA (Pal et al., 2022).

**Baselines.** We select three sets of baselines: 1) *PEFT baselines*: including a) Full Fine-tuning (FT), b) Houlsby Adapter, c) Vanilla LoRA, and d) Prompt Tuning. 2) *Multi-task learning baselines*: selecting advanced multi-task frameworks, including MultiLoRA (Wang et al., 2023a), MixLoRA (Li et al., 2024), HydraLoRA (Tian et al., 2024), and MTL-LoRA (Yang et al., 2025b).3) *Continual learning baselines*:ILoRA (Li et al., 2025) and EWC (Kirkpatrick et al., 2017). All results are averaged over three independent runs with different random seeds.

**Implementation.** We use T5-Base, Qwen3-4B, and Qwen3-8B as backbones. T5-Base is trained with a batch size of 64 and a learning rate of $3 \times 10^{-4}$. Qwen uses a batch size of 8 with 4-step gradient accumulation and a learning rate of $2 \times 10^{-5}$. All models are optimized using AdamW with a weight decay of 0.01, a cosine decay schedule, 1000-step warm-up, and 20,000 total steps.The input sequence length is capped at 128 tokens. Training is performed on NVIDIA RTX 5880 Ada Generation. For fair comparison, all models share a unified training pipeline and identical data sampling strategies. Further details on the datasets and implementation are provided in Appendix A and Appendix B.

Table 1: GLUE results with T5-base. We report Accuracy (SST-2, MNLI, QNLI, QQP, RTE, MRPC), MCC (CoLA), and Pearson correlation (STS-B). Scores are percentages rounded to one decimal, with best results in **bold**.

| Method | RTE | MNLI | MRPC | SST2 | QQP | QNLI | CoLA | STSB | AVG. |
|---|---|---|---|---|---|---|---|---|---|
| Adapter | 75.1 | 85.0 | 88.2 | 93.7 | 90.3 | 92.8 | 59.7 | 91.2 | 84.5 |
| PromptTuning | 65.0 | 85.4 | 87.2 | 92.1 | 88.8 | 88.8 | 38.6 | 89.0 | 79.4 |
| $LoRA_{r=8}$ | 75.5 | 86.2 | 88.4 | 93.6 | 89.8 | 93.3 | 64.1 | 92.2 | 85.4 |
| $LoRA_{r=16}$ | 79.8 | 86.0 | 87.7 | 94.0 | 89.8 | 93.0 | 61.1 | 91.8 | 85.4 |
| Finetuning | 75.5 | 85.8 | **89.0** | **94.3** | 89.8 | 92.4 | 60.7 | 91.7 | 84.9 |
| HydraLoRA | 79.4 | 86.0 | 87.0 | 94.2 | 90.0 | 93.1 | 62.2 | 91.7 | 85.6 |
| MTL-LoRA | 75.8 | **86.4** | 88.2 | 93.6 | **90.2** | **93.4** | 64.1 | 92.5 | 85.5 |
| MixLoRA | 76.9 | 86.1 | 87.9 | 93.7 | 90.0 | 92.9 | **65.6** | 92.3 | 85.7 |
| MultiLoRA | 76.5 | 86.1 | 88.4 | 93.9 | 89.8 | 93.1 | **65.6** | **92.8** | 85.8 |
| MoSE | **80.9** | 86.2 | 88.7 | 93.5 | **90.2** | 93.2 | **65.6** | 92.2 | **86.3** |

## 4.2 MULTI-TASK LEARNING AND TRANSFERRING

Table 1 summarizes results on the GLUE benchmark, covering tasks of varying sizes. We can observe that *MoSE* achieves the highest average performance and excels on low-resource tasks such as RTE (80.9%), QQP (90.2%), and CoLA (65.6%). This result demonstrates *MoSE*'s superior ability to retain task-specific knowledge, leading to stronger fine-tuning performance. Compared to strong multi-task baselines (e.g., MultiLoRA and MixLoRA), *MoSE* consistently delivers gains across tasks, highlighting its robustness in multi-task settings.

For commonsense reasoning, Table 2 shows that *MoSE* consistently outperforms all baselines across different backbones. Notably, it achieves substantial gains on harder tasks like HellaSwag, SIQA, and OBQA—for example, a 6.8% improvement on HellaSwag over LoRA. These tasks demand nuanced reasoning and deeper context understanding, where *MoSE*'s structured use of shared and routing experts enables effective knowledge sharing and task-specific adaptation, leading to superior multi-task performance.

Table 2: Results across commonsense reasoning datasets under different model backbones. All values are accuracy scores (%). The best results are in **bold**.

| Method | BoolQ | PIQA | SIQA | HellaS. | WinoG. | ARC-E | ARC-C | OBQA | AVG. |
|---|---|---|---|---|---|---|---|---|---|
| **Qwen3-4B** | | | | | | | | | |
| LoRA | 83.9 | 80.3 | 73.0 | 80.6 | 72.3 | 92.6 | 84.6 | 84.8 | 81.5 |
| MultiLoRA | 77.4 | 80.6 | 71.0 | 80.0 | 69.5 | 94.1 | 87.0 | 83.3 | 80.4 |
| MixLoRA | 82.4 | 81.5 | 74.2 | 81.5 | 69.4 | 94.4 | 86.6 | 85.2 | 81.9 |
| HydraLoRA | 82.5 | 80.2 | 73.4 | 83.7 | 69.7 | 93.6 | 86.8 | 85.8 | 82.0 |
| MTL-LoRA | 85.4 | 84.3 | **76.4** | 83.5 | 74.4 | 94.5 | **87.3** | 86.3 | 84.0 |
| MoSE | **86.1** | **84.9** | 76.1 | **85.3** | **77.7** | **94.7** | 86.5 | **86.7** | **84.8** |
| **Qwen3-8B** | | | | | | | | | |
| LoRA | 82.8 | 85.6 | 75.2 | 81.7 | 78.3 | 96.3 | 89.9 | 88.0 | 84.7 |
| MultiLoRA | 81.5 | 84.6 | 74.8 | 82.6 | 73.7 | 96.2 | **90.7** | 87.3 | 83.9 |
| MixLoRA | 83.8 | 85.4 | 75.7 | 87.2 | 77.1 | 96.3 | 90.6 | 89.2 | 85.7 |
| HydraLoRA | 84.3 | 87.4 | 75.9 | 86.1 | 78.9 | 95.8 | 89.9 | 88.2 | 85.8 |
| MTL-LoRA | 83.2 | 85.4 | 77.5 | 84.7 | 74.7 | **96.4** | 90.4 | 87.6 | 85.0 |
| MoSE | **87.2** | **88.2** | **79.9** | **88.5** | **81.1** | 95.8 | **90.7** | **89.4** | **87.6** |

Moreover, we evaluate transferability to assess how well each model captures commonsense reasoning knowledge. As shown in Table 3, *MoSE* achieves an average accuracy of 69.9% across four reasoning tasks in a zero-shot setting, demonstrating strong generalization. In contrast, existing multi-task baselines show substantial variance—for example, HydraLoRA performs well on CSQA but poorly on CSQA2 and QASC, likely due to overfitting to source tasks. Compared with MultiLoRA and MixLoRA, *MoSE* consistently delivers stronger transfer performance, suggesting that its shared-expert mechanism and task-aware routing work effectively together to capture shared knowledge. This design also equips *MoSE* with stronger resistance to forgetting, which we further validate in the subsequent continual learning experiments.

Table 3: Performance on additional reasoning benchmarks after fine-tuning on commonsense tasks using Qwen3-4B. All values are accuracy scores (%).

| Method | CSQA | CSQA2 | QASC | LogiQA | AVG. |
|---|---|---|---|---|---|
| LoRA | 45.9 | 23.6 | 35.1 | 6.6 | 27.8 |
| MultiLoRA | 70.8 | 60.8 | 94.8 | 42.1 | 67.1 |
| MixLoRA | 73.8 | **62.2** | 95.3 | 43.6 | 68.0 |
| HydraLoRA | 66.3 | 37.2 | 35.9 | 34.7 | 43.5 |
| MoSE | **76.1** | 61.4 | **95.9** | **46.4** | **69.9** |
| *Compared to LoRA* | +30.2 | +37.8 | +60.8 | +39.8 | +42.1 |

In summary, by combining a dedicated shared expert, stabilized by sparse updates with flexible routing experts, *MoSE* builds a robust general foundation that excels at both broad linguistic tasks and deep, nuanced reasoning. The effectiveness and scalability of *MoSE* are further confirmed by its consistent gains across multiple model sizes.

### 4.3 Continuing Learning and Forgetting Resistance

As discussed in Section 1, MoE experts should capture both shared and task-specific knowledge to enhance multi-task generalization. To validate this, we thoroughly evaluate *MoSE*'s continual learning and forgetting resistance.

We first simulate a continual multi-task setting where models are first fine-tuned on commonsense reasoning tasks, then adapted to QA (CSQA, CSQA2, LogiQA), inference (SciTail), and reading comprehension (RACE). As shown in Table 4, *MoSE* achieves the lowest forgetting score, demonstrating strong knowledge retention. Remarkably,during continual fine-tuning on QA tasks, MixLoRA forgets 34.2% on the initial reasoning tasks, while *MoSE* drops only 5.3%, highlighting its resilience to forgetting.

Table 4: Continual learning results. We report initial accuracy (**Init**), final accuracy (**Final**), forgetting (**Forget**), and target-task accuracy (**Target**).

| Backbone | Method | Init | QAs | | | SciTail | | | RACE | | |
|---|---|---|---|---|---|---|---|---|---|---|---|
| | | | Final | Forget | Target | Final | Forget | Target | Final | Forget | Target |
| | MultiLoRA | 80.4 | 65.9 | −14.5 | 58.6 | 75.7 | −4.7 | 91.5 | 73.4 | −7.0 | 85.9 |
| Qwen3-4B | MixLoRA | 82.2 | 48.0 | −34.2 | 62.3 | 78.8 | −3.4 | **96.4** | 47.2 | −35.0 | 86.1 |
| | MoSE | **84.8** | **79.5** | **−7.0** | 63.7 | **83.6** | **−1.2** | 95.1 | **83.5** | **−1.3** | 86.3 |
| | MultiLoRA | 83.9 | 79.1 | −4.8 | 60.1 | 82.1 | −1.8 | 88.9 | 81.3 | −2.6 | 90.3 |
| Qwen3-8B | MixLoRA | 85.7 | 78.2 | −7.5 | 64.7 | 85.2 | −0.5 | 94.8 | 84.0 | −1.7 | 91.0 |
| | MoSE | **87.6** | **84.5** | **−3.1** | 67.2 | **87.3** | **−0.3** | 95.2 | **87.0** | **−0.6** | 91.1 |

We further evaluate continual learning performance under a sequential task setting and report two CL metrics— backward transfer (Lopez-Paz & Ranzato, 2017) and forgetting score.The corresponding results are summarized in Table 5. *MoSE* consistently achieves the strongest performance, surpassing MultiLoRA and MixLoRA by 5.1% and 28.7% on the initial reasoning tasks, respectively. In contrast, MixLoRA suffers from pronounced forgetting, with a 57.1% accuracy drop on reasoning datasets, highlighting its instability. Overall, these results indicate that *MoSE* better preserves common knowledge while adapting to new tasks, aligning with our central goal of enabling effective knowledge sharing and task-specific retention in multi-task fine-tuning. In addition,more details about the metrics and the full continual learning results are provided in Appendix E.

Table 5: Sequential continual learning performance on Qwen3-8B, reported as accuracy (%).(Reason → QAs → RACE → SciTail → FOMC → MedMCQA).

| Method | Reason | QAs | RACE | SciTail | FOMC | MedMCQA | Avg. | Forget.↓ | Backward.↑ |
|---|---|---|---|---|---|---|---|---|---|
| MoSE | **85.3** | **57.3** | **90.3** | 91.4 | **65.7** | **57.7** | **74.6** | **3.0** | **-3.0** |
| MixLoRA | 50.2 | 53.8 | 89.1 | 19.8 | 59.1 | 52.2 | 54.0 | 28.7 | -26.0 |
| MultiLoRA | 77.9 | 52.4 | 88.1 | **91.5** | 55.6 | 52.6 | 69.7 | 3.8 | -3.7 |
| ILoRA | 70.4 | 49.5 | 88.7 | 78.2 | 44.6 | 50.7 | 63.7 | **3.0** | -4.1 |
| EWC | 44.9 | 38.4 | 74.8 | 77.8 | 48.6 | 44.8 | 54.9 | 13.28 | -16.2 |

### 4.4 Capability of Shared and Routing Experts

For a more comprehensive examination of our expert design and update strategies, we conduct the following experiments to provide an in-depth analysis of the proposed *MoSE*.

**Effectiveness of Shared Experts.** We first examine the impact of different sparse update ratios in Figure 3(a), where we observe that updating only 5% of parameters yields the best performance. This demonstrates that a small, carefully selected subset of updates is sufficient for capturing shared knowledge while mitigating forgetting. Furthermore, Figure 3(b) visualizes the update frequency of shared experts during training, where brighter stripes correspond to more frequently updated regions. The structured patterns indicate that our sparse update mechanism consistently targets meaningful shared parameters rather than relying on random or noisy updates.

**Effectiveness of Routing Experts.** For routing experts, we aim for clear specialization across tasks, encouraging each expert to capture task-specific information. Thus, we visualize the task embedding and routing expert activation on Figure 4. As shown in Figure 4(a), embeddings from the same task

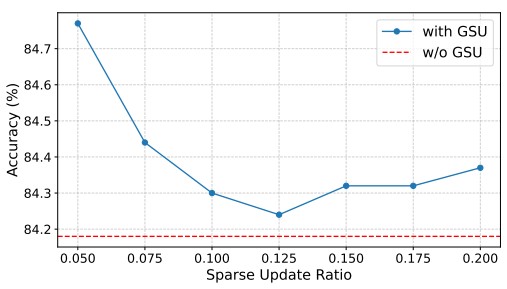
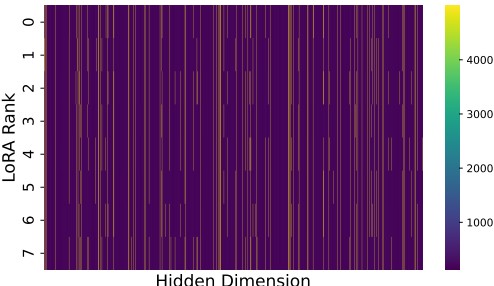

(a) Sensitivity to sparse update ratio.  (b) Update frequency heatmap of shared expert.

Figure 3: Analysis of GSU(Gradient-based Sparse Update).

form tight, well-separated clusters, suggesting that the router input effectively encodes task identity. In Figure 4(b), we observe that each task activates a distinct subset of experts, with similar tasks (e.g., ARC-E and ARC-C) often selecting overlapping experts (e.g., experts 5, 6, and 8), reflecting shared characteristics. These patterns demonstrate that our routing strategy successfully captures both task-specific and shared structures. Additionally, due to space constraints, further analyses of the shared experts and routing experts are provided in Appendix D.2 and Appendix D.1.

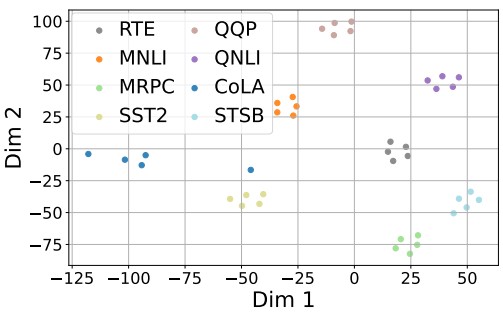
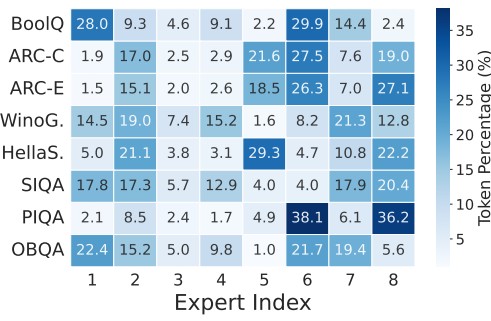

(a) Task embedding visualization.  (b) Token-level expert routing distribution.

Figure 4: Task-level diversity and expert routing.

**Ablation Study.** The above experiments demonstrate the superiority of *MoSE*. We further investigate the contribution of each component through an ablation study (Table 6). Removing gradient-based sparse updates (w/o GSU) causes the largest drop—especially on reasoning tasks—highlighting their role in preserving prior knowledge and mitigating interference. The shared expert (w/o Shared) and Top-$k$ selection (vs. w/random) also prove essential. Meanwhile, FiLM removal yields moderate declines, reflecting its task-aware modulation, and omitting the orthogonality constraint reduces accuracy, underscoring its benefit in disentangling shared and task-specific features. Together, these results confirm that all components are integral to *MoSE*'s performance.

In addition, ablation results on applying GSU and task-aware routing to MixLoRA and MultiLoRA, as well as further analyses under increased parameter budgets, are provided in Appendix D.3. These results further support the rationale behind incorporating these techniques into our decoupled design.

Table 6: Ablation study on different components of *MoSE* across two settings: T5-base on GLUE and Qwen3-4B on reasoning datasets.

| Method | T5-GLUE | Qwen3-Reason |
|---|---|---|
| MoSE | **86.3** | **84.8** |
| w/o FiLM | 85.7 | 84.4 |
| w/o Orth | 85.7 | 84.5 |
| w/o GSU | 85.9 | 84.1 |
| w/o Shared | 85.7 | 84.2 |
| w/ Random | 85.9 | 84.0 |

**Analysis of Efficiency.** Table 7 summarizes the structural configurations, parameter budgets, memory usage, and training time of all methods. Dense MoE-LoRA variants such as MultiLoRA and Hy-

Table 7: Comparison of expert rank, number of experts, type, trainable parameter ratio (TP%), memory usage, training time, and performance across methods on Qwen.

| Method | Rank | Experts | Type | TP (%) | Memory | Time | MTL | CL |
|--------|------|---------|------|--------|--------|------|-----|-----|
| MultiLoRA | 24 | 3 | Dense | 4.00 | 46.9G | 10.2h | 83.9 | 69.7 |
| MixLoRA | 8 | 8 | Sparse | 2.62 | 41.0G | 17.3h | 85.7 | 54.0 |
| HydraLoRA | 24 | 3 | Dense | 2.82 | 46.0G | 9.3h | 85.8 | – |
| MTL-LoRA | 24 | 3 | Dense | 2.76 | 45.7G | 10.3h | 85.0 | – |
| MoSE | 8 | 9 | Sparse | 2.94 | 45.3G | 18.1h | 87.6 | 74.6 |

draLoRA activate all experts during training, achieving faster optimization through high parallelism but incurring substantially higher peak memory. In contrast, *MoSE* employs shared experts and task-aware routing, adding only modest parameter and computational overhead relative to MixLoRA while delivering significantly better multi-task accuracy and forgetting resistance. Although sparse-routing MoE-LoRA methods naturally introduce extra memory and computation—and *MoSE* is no exception—the gains observed in the main results show that this overhead is both reasonable and justified. Additional details on trainable parameters are provided in Appendix D.4.

**Analysis of Backward Transfer.** Table 8 reports the backward transfer results across sequential tasks. Backward transfer measures the gap between a task's final performance and its initial performance when it was first learned, capturing how much later tasks interfere with previously acquired knowledge. *MoSE* achieves the highest average BWT ($-3.0$), reflecting the smallest performance degradation when revisiting earlier tasks. It consistently limits negative transfer across all benchmarks, particularly on Reason and SciTail, where competing methods such as MixLoRA and EWC suffer substantial drops. This suggests that *MoSE*'s decoupled shared–expert structure mitigates interference from later tasks, enabling stable cross-task adaptation without overwriting initial task representations. Overall, *MoSE* exhibits strong robustness to backward interference and maintains more favorable transfer dynamics than all baselines.

Table 8: Backward transfer (BWT) performance on Qwen3-8B (higher is better).

| Method | Reason | QAs | RACE | SciTail | FOMC | Avg. |
|--------|--------|-----|------|---------|------|------|
| MoSE | -2.3 | -8.7 | -0.2 | -3.4 | -0.4 | **-3.0** |
| MixLoRA | -35.5 | -10.9 | -1.5 | -75.9 | -6.2 | -26.0 |
| MultiLoRA | -6.0 | -7.7 | -0.9 | -0.6 | -3.1 | -3.7 |
| ILoRA | -8.0 | -3.6 | 0.1 | -4.4 | -4.6 | -4.1 |
| EWC | -39.8 | -15.2 | -7.0 | -7.6 | -11.5 | -16.2 |

## 5 CONCLUSION

To tackle the cataclysmic forgetting problem during multi-task fine-tuning, we proposed *MoSE*, a novel MoE-based PEFT framework that explicitly separates the learning process of shared and task-specific knowledge. Specifically, we first divided experts into shared experts and routing experts. Then, for routing experts, we employed FiLM-based task-aware modulation to train router to select the appropriate routing experts for task-specific knowledge learning, which were updated with entire expert module. For shared experts, we developed a Top-$k$-selection strategy to selectively update part of parameters in shared experts, which is helpful for memorizing as much common knowledge as possible. Meanwhile, an orthogonality constraint together with an expert load-balancing mechanism was introduced to further stabilize the tuning process. Extensive evaluations on multi-task, transfer, and continual learning benchmarks demonstrated that *MoSE* consistently outperforms advanced PEFT and multi-task adaptation methods, underscoring its potential as a scalable and robust solution for dynamic, heterogeneous task environments. In the future, we plan to extend the applications of *MoSE*, and consider more efficient tuning strategies for larger models.

# 6 STATEMENT

## 6.1 ETHIC STATEMENT

This work adheres to the ICLR Code of Ethics. No human subjects or animal experimentation were involved in this study. All datasets used in our experiments (e.g., GLUE, CSQA, RACE, SciTail) are publicly available and widely adopted in prior research, and were utilized in full compliance with their respective licenses and usage guidelines. No personally identifiable or sensitive data were used. We have taken care to avoid introducing bias or discriminatory outcomes in our methodology and evaluation. We do not anticipate any ethical concerns or negative societal impacts arising from this study. We are committed to ensuring transparency, fairness, and integrity throughout the entire research process.

## 6.2 REPRODUCIBILITY STATEMENT

To ensure reproducibility, we clearly document the experimental configurations across both the main text and the appendix. In Section 4.1, we provide the primary setup, including evaluation benchmarks, selected baselines, backbone models, and training pipelines. Key hyperparameters such as batch sizes, learning rates, optimizer settings, warm-up steps, and total training steps are also reported to enable faithful replication.

Further details are provided in the appendix. Appendix A describes the datasets used in multi-task learning, transfer, and continual learning, along with their splits and evaluation protocols. All datasets used are publicly available. Appendix B expands on model-specific configurations, including *MoSE* architecture choices, LoRA rank adjustments, and continual learning schedules. Appendix C details the balanced data sampling strategy, designed to account for variations in dataset sizes and ensure fair training across tasks. Together, these sections provide comprehensive coverage of all implementation and experimental details.

In addition, we provide our code at the following link[1] and also include it in the supplementary materials, which will facilitate independent verification and further research.

---

[1] https://anonymous.4open.science/r/MoSE-F2BD/README.md

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

# A  DATASETS

## A.1  MTL BENCHMARK

**GLUE Benchmark.** We utilize the GLUE benchmark (Wang et al., 2018) to evaluate the generalization capability of our model across a diverse set of natural language understanding tasks. GLUE covers a broad spectrum of tasks, including paraphrase detection (MRPC, QQP), sentiment classification (SST-2), natural language inference (MNLI, RTE, QNLI), linguistic acceptability (CoLA), and semantic textual similarity (STS-B). Following prior work (Wang et al., 2023b), for tasks with fewer than 10,000 training examples (i.e., MRPC, STS-B, and CoLA), we evenly split the original validation set into new validation and test sets to enable consistent evaluation.

**Commonsense Reasoning.** For commonsense reasoning tasks, following prior work (Liu et al., 2024d; Yang et al., 2025b; Li et al., 2024), we adopt a benchmark suite consisting of eight widely-used datasets: BoolQ, PIQA, SIQA, HellaSwag, WinoGrande, ARC-Easy, ARC-Challenge, and OpenBookQA. These tasks span multiple reasoning paradigms, including question answering based on implicit facts (BoolQ), physical commonsense (PIQA), social scenarios (SIQA), script and event plausibility (HellaSwag), coreference-based reasoning (WinoGrande), elementary and challenging science exams (ARC-E and ARC-C), and open-domain factual reasoning (OBQA). Together, they provide a comprehensive evaluation of a model's capability to reason under various commonsense contexts. For GLUE benchmark and commonsense reasoning tasks, we selected the checkpoint with the highest average performance on validation set.

## A.2  TRANSFERRING AND CONTINUAL LEARNING

**Transferring.** For transferability evaluation, we introduce four commonsense question answering benchmarks: CommonsenseQA (CSQA) (Talmor et al., 2019), CommonsenseQA 2.0 (CSQA2) (Talmor et al., 2021), LogiQA(Liu et al., 2021), and QASC(Khot et al., 2020). While all four tasks fall under the umbrella of commonsense QA, they differ from the previously used reasoning benchmarks in both domain and task formulation. Specifically, CSQA and CSQA2 emphasize multi-hop reasoning over structured knowledge; LogiQA focuses on logical consistency within reading comprehension passages; and QASC requires compositional reasoning by combining facts across multiple sentences. This domain shift presents a more rigorous test of the model's zero-shot generalization and transfer capabilities.

**Continual Learning.** In the continual learning setting, we construct a sequence of five task groups to evaluate both forgetting resistance and adaptability. The first group comprises CSQA, CSQA2, and LogiQA, representing commonsense question answering. The second task is SciTail (Khot et al., 2018), a natural language inference benchmark. Next, the model learns RACE (Lai et al., 2017), a reading comprehension dataset that introduces longer-context reasoning demands. After RACE, we further include FOMC (Shah et al., 2023), a financial reasoning benchmark, and MedMCQA(Pal et al., 2022), a medical-domain multiple-choice QA dataset. This extended and diverse task sequence introduces progressively shifting domains—commonsense, inference, long-form comprehension, finance, and medicine—thereby simulating realistic task drift and rigorously testing the model's ability to retain prior knowledge while continually acquiring new skills.

# B  IMPLEMENTATION

## B.1  GENERAL SETTINGS

We use T5-Base, Qwen3-4B, and Qwen3-8B as backbone models. T5-Base is trained with a batch size of 64 and a learning rate of $3 \times 10^{-4}$, while Qwen models adopt a batch size of 8 with 4-step gradient accumulation and a learning rate of $2 \times 10^{-5}$. All models are optimized with AdamW using a weight decay of 0.01, a cosine learning rate schedule, 1,000 warm-up steps, and 20,000 total training steps. The maximum input sequence length is set to 128 tokens. Training is performed on NVIDIA RTX 5880 Ada Generation GPUs. To reduce memory and computation overhead, we apply uniform 4-bit NF4 quantization with double quantization and employ FP16 precision for training.

## B.2 Detailed Settings

***MoSE*** **Configuration.** Our *MoSE* architecture comprises one shared expert and eight routing experts, each with rank 8 and scaling factor 16. For each input, the router selects the top-2 routing experts along with the shared expert. Task embeddings are obtained by averaging the T5 encoder representations of 20 randomly sampled training examples, yielding a 768-dimensional vector. The FiLM modulation layer uses a compression rank of 32 to limit parameter overhead. For sparse updates, we select the top 5% most important parameters . We set the EMA momentum coefficient to 0.9, apply a balanced load constraint with weight 0.002, and an orthogonality loss with weight 0.01. The *MoSE* module is applied only to feed-forward network (FFN) layers, while attention layers use standard single-rank LoRA adapters.

**Baselines Configuration.** All baseline models are implemented under the same training and data pipeline to ensure fair comparison. To maintain comparable parameter budgets, we adjust the LoRA rank for each method. For instance, in MultiLoRA we employ three experts with rank 24 to match the total parameter count of our approach. The number of experts and their corresponding ranks for all methods are reported in Table 7.

## B.3 Continual Learning Settings

During continual learning, we use a small learning rate of $5 \times 10^{-6}$ to help the model acquire new knowledge without forgetting prior information. Fine-tuning is performed for 6000 steps on QA tasks (CSQA, CSQA2, LogiQA) and 2000 steps on SciTail,RACE,FOMC and MedMCQA. Evaluation is conducted at the final training step. For RACE, we increase the maximum input length to 256 tokens to handle longer passages.For fairness, all methods are evaluated under the same protocol, and **no data replay** is employed for any model.

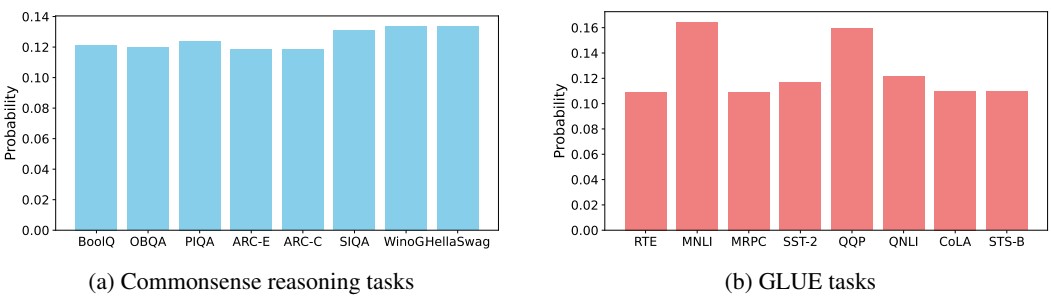

(a) Commonsense reasoning tasks          (b) GLUE tasks

Figure 5: Sampling probabilities across different task groups.

## C Balanced Data Sampling

In multi-task settings, task datasets often differ significantly in size. Uniform sampling thus risks overfitting large tasks while undertraining smaller ones, impairing generalization. To address this, we adopt a weighted sampling strategy that adjusts task frequencies without explicit resampling.

Specifically, given $T$ task datasets $\{\mathcal{D}_1, \mathcal{D}_2, \ldots, \mathcal{D}_T\}$ with sizes $\{n_1, n_2, \ldots, n_T\}$, we compute task-level sampling probabilities using a softmax over dataset sizes:

$$p_i = \frac{\exp(n_i)}{\sum j = 1^T \exp(n_j)}. \tag{11}$$

Then, we assign all samples from task $\mathcal{D}i$ an equal per-sample weight for uniform treatment:

$$w_{ij} = \frac{p_i}{n_i}, \quad j = 1, \ldots, n_i. \tag{12}$$

This strategy mitigates inter-task imbalance while ensuring intra-task fairness. Since sampling is global, each batch naturally contains a mix of tasks, facilitating balanced supervision and knowledge sharing. The sampling probabilities for different datasets are illustrated in Figure 5. They remain approximately balanced across sub-tasks, preventing excessive bias toward or neglect of any single task. While this scheme may not constitute the optimal sampling strategy, it is uniformly applied to all methods to ensure comparability and fairness.

# D SUPPLEMENTARY ANALYSIS OF *MoSE*

## D.1 ANALYSIS OF ROUTING EXPERTS

**Router Input Strategy Analysis.** To examine the impact of routing inputs and modulation strategies, we compare two alternatives: *Task-level Routing* and *Token-level Routing*, each evaluated with and without Feature-wise Linear Modulation (FiLM). As shown in Table 9, *Token-level Routing* with FiLM yields the best performance (86.3), suggesting that conditioning the router on richer contextual signals is particularly beneficial when combined with modulation. In contrast, for *Task-level Routing*, applying FiLM slightly reduces performance (85.7 vs. 85.8), indicating that task embeddings may already encode sufficient task-specific information and that additional modulation could introduce noise. Overall, these results underscore the importance of input representation choices and reveal the nuanced role of FiLM in designing effective routing strategies for MoE architectures.

Table 9: Comparison of Task-level and Token-level Routing strategies on T5-Base with the GLUE benchmark, under both FiLM and non-FiLM settings.

| Routing Strategy | With FiLM | Without FiLM |
|---|---|---|
| Task-level Routing | 85.7 | 85.8 |
| Token-level Routing | 86.3 | 85.7 |

**Expert Load Balancing.** To mitigate expert under-utilization, we apply a load-balancing constraint with a small weight($\lambda = 0.002$) during training. As shown in Figure 6a, the cumulative number of activated tokens per expert demonstrates a relatively balanced distribution. Despite minor variations, the load is relatively balanced across experts, with no signs of collapse or severe skew. This mild imbalance reflects our router's specialization capability, allowing different experts to focus on diverse input patterns without compromising overall utilization.

**Task Embedding Visualization.** We also visualize the task embeddings of reasoning datasets to examine whether our model captures meaningful task-level distinctions. As illustrated in the middle plot of Figure 6b, embeddings from the same task are tightly clustered, while those from different tasks are well-separated. Interestingly, semantically similar tasks—such as ARC-C and ARC-E, or PIQA and SIQA—appear close to each other in the embedding space. This suggests that our task encoder not only preserves task-specific distinctions but also reveals latent task-level relationships.

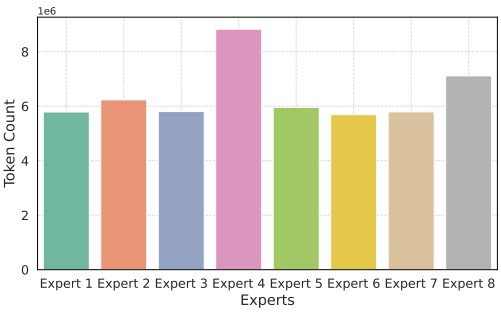
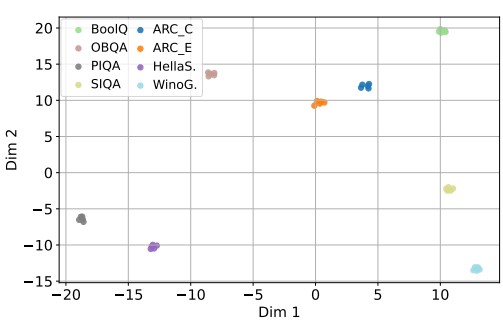

(a) Expert token count.

(b) Task embedding visualization.

Figure 6: Visual analysis of *MoSE*'s internal behavior: (a) shows load balance across experts, and (b) displays the separability of task embeddings.

## D.2 ANALYSIS OF SHARED EXPERT

**Analysis of Orthogonality Loss Weight.** To better understand the impact of orthogonality constrain on model performance, we conduct ablation experiments with different regularization weights. As shown in Figure 7a, small regularization weights (e.g., 0.001 and 0.005) bring noticeable improvements over the baseline without orthogonal loss, with the best performance achieved at $\beta = 0.01$. However, excessively large weights (e.g., 0.05 and 0.1) lead to performance degradation. These re-

sults suggest that a moderate orthogonal constraint encourages diverse expert representations, while overly strong regularization hinders optimization and hurts generalization.

**Shared Expert Weight Analysis.** In addition, we examine the activation weights of the shared expert, as shown in Figure 7b. Consistent with its role as a lower-layer–like general feature extractor, the shared expert typically receives smaller activation weights, indicating that it does not dominate the final prediction. Instead, it provides a stable foundational representation that supports cross-task knowledge sharing, while the routed experts—analogous to upper-layer task heads—carry larger weights to capture task-specific signals and drive the final decisions.

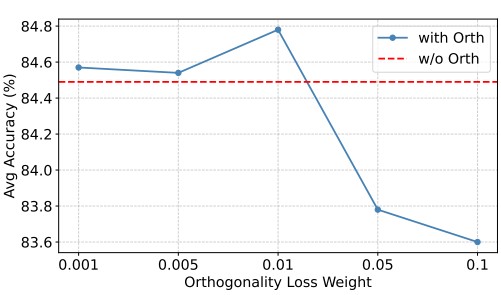
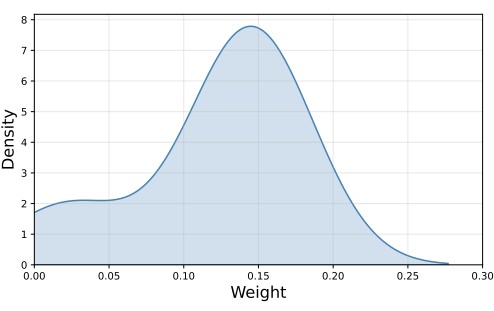

(a) Effect of orthogonality loss weight.

(b) Weight distribution of the shared expert.

Figure 7: Visual analysis of *MoSE*: (a) sensitivity to orthogonality loss weight, and (b) distribution of shared expert weights.

### D.3 SUPPLEMENTARY ABLATION STUDY

**Ablation on MultiLoRA and MixLoRA.** *MoSE* can be viewed as a generalized framework that unifies characteristics of both dense and sparse MoE architectures. Consequently, performing ablations on MultiLoRA (dense) and MixLoRA (sparse) is both necessary and well-justified. To this end, we further conduct fine-grained component-level ablations by adding either task-aware routing or gradient-based sparse updates on top of each baseline. The results are reported in Table 10.

Table 10: Ablation study on routing, FiLM modulation, and sparse GSU updates.

| Method | Avg. | Compare to base |
|---|---|---|
| base MultiLoRA | 80.4 | \ |
| base MixLoRA | 81.9 | \ |
| MultiLoRA + Routing | 82.4 | +2.0 |
| MultiLoRA + FiLM | 83.3 | +2.9 |
| MixLoRA + FiLM | 82.4 | +0.5 |
| MultiLoRA + GSU | 65.6 | -14.8 |
| MixLoRA + GSU | 80.1 | -1.8 |
| MoSE | 84.7 | \ |

The results show that task-aware routing consistently improves performance, regardless of whether the base architecture is MultiLoRA or MixLoRA. In contrast, adding gradient-based sparse updates (GSU) alone is not well-suited to either dense or sparse MoE-LoRA structures, and even leads to substantial degradation when applied to MultiLoRA. This indicates that GSU must be paired with a decoupled design—where shared and routing experts play distinct, complementary roles—in order to fully realize its stability and interference-resistance benefits.

**Ablation on Trainable Parameters.** To verify that the performance gains of *MoSE* do not arise merely from adding more architectural components or increasing parameter counts, we conduct a series of ablations on MixLoRA. As a lightweight sparse MoE-LoRA baseline, MixLoRA can be regarded as the foundational variant from which *MoSE* is extended. Specifically, we augment MixLoRA with a task-aware FiLM module, add an additional expert, and double the rank of the LoRA adapters.

Table 11: Ablation on different MixLoRA variants and their trainable parameter (TP) ratios.

| Method | TP | Avg. |
|---|---|---|
| MoSE | 4.63% | 84.7 |
| MixLoRA | 4.13% | 81.9 |
| MixLoRA + FiLM | 4.17% | 82.4 |
| MixLoRA + expert | 4.60% | 81.3 |
| MixLoRA with double rank | 7.77% | 82.5 |

As shown in Table 11, simply increasing the number of trainable parameters does not yield meaningful performance gains; in fact, adding an extra expert even leads to a noticeable drop in accuracy. This suggests that a larger parameter budget can introduce additional optimization challenges and gradient noise, and does not inherently improve the model's expressive capacity. Consequently, the performance improvements of *MoSE* cannot be attributed to increased parameter count alone, but instead arise from its structural design—specifically, the combination of task-aware routing and gradient-based sparse updates.

**Freezing the Shared Expert.** In our continual learning (CL) experiments, we additionally evaluate a "frozen shared expert" variant, in which the shared expert is trained only during the first task group (Reason tasks) and kept fixed thereafter. The results, shown in Table 12, indicate that although this configuration still achieves reasonable overall performance, it is clearly inferior to the full *MoSE* model. This further highlights the importance of keeping the shared expert adaptable throughout training and demonstrates the effectiveness of *MoSE*'s shared-expert design.

Table 12: Effect of freezing the shared expert in continual learning.

| Method | Reason | QAs | RACE | SciTail | Avg. |
|---|---|---|---|---|---|
| MoSE | 85.8 | 65.6 | 91.0 | 94.8 | **84.3** |
| Freezing | 84.3 | 58.3 | 88.9 | 88.6 | 80.0 |

### D.4 TRAINABLE PARAMETER ANALYSIS

**Parameter Breakdown of Trainable Components.** We present a parameter breakdown of the *MoSE* framework on Qwen3-8B in Table 13. Compared to prior MoELoRA-based architectures (Li et al., 2024; Gao et al., 2024), *MoSE* introduces two additional components—the Shared Expert and the FiLM Adapter—which together account for only 0.31% of the total parameters. Despite this negligible increase, *MoSE* delivers strong multi-task performance and demonstrates remarkable resistance to forgetting, highlighting the effectiveness of our design in balancing parameter efficiency, task generalization, and long-term knowledge retention.

Table 13: Parameter breakdown of trainable components in *MoSE* based on Qwen3-8B. Trainable Ratio and Total Ratio indicate the percentage of trainable and total model parameters, respectively.

| Component | Params | Trainable Ratio | Total Ratio |
|---|---|---|---|
| Single LoRA | 7.67M | 5.37 | 0.16 |
| Routing Experts | 113.25M | 79.28 | 2.33 |
| Shared Expert | 14.16M | 9.91 | 0.29 |
| Router FC | 6.64M | 4.65 | 0.14 |
| FiLM Adapter | 1.13M | 0.79 | 0.02 |
| **Total Trainable** | **142.84M** | **100.00** | **2.94** |

**Parameter Efficiency and Performance Comparison of T5.** In addition, we present a comparison of parameter ratios and final performance across different methods, as summarized in Table 14. *MoSE* achieves the highest GLUE performance (86.3) while maintaining a comparable trainable parameter ratio (6.76%). Relative to other PEFT methods such as LoRA (85.4, 2.83%) and MultiLoRA (85.8, 6.67%), *MoSE* provides consistent improvements without introducing significant

parameter overhead. These findings highlight that our proposed architecture strikes a favorable balance between efficiency and effectiveness.

Table 14: Comparison of parameter efficiency (TP, ratio of trainable parameters to total parameters) and average GLUE performance on T5-Base across different methods.

| Method | Adapters | PT | FT | LoRA | HydraLoRA | MTL-LoRA | MixLoRA | MultiLoRA | MoSE |
|---|---|---|---|---|---|---|---|---|---|
| TP (%) | 5.99 | 0.01 | 100 | 2.83 | 5.74 | 5.66 | 5.80 | 6.67 | 6.76 |
| Perform. | 84.5 | 79.4 | 84.9 | 85.4 | 85.6 | 85.5 | 85.7 | 85.8 | 86.3 |

# E    SUPPLEMENTARY ANALYSIS OF CONTINUAL LEARNING

## E.1    DETAILED PERFORMANCE

Figure 8 illustrates the accuracy dynamics of both QA and reasoning tasks during continual fine-tuning on QA tasks. *MoSE* begins with stronger QA performance and continues to improve steadily, while its accuracy on previously learned reasoning tasks shows an initial drop followed by rapid recovery, indicating robust knowledge retention. In contrast, MultiLoRA and MixLoRA suffer from severe forgetting, with reasoning performance steadily declining. This highlights the effectiveness of *MoSE*'s expert design and update strategy in balancing adaptation and stability.

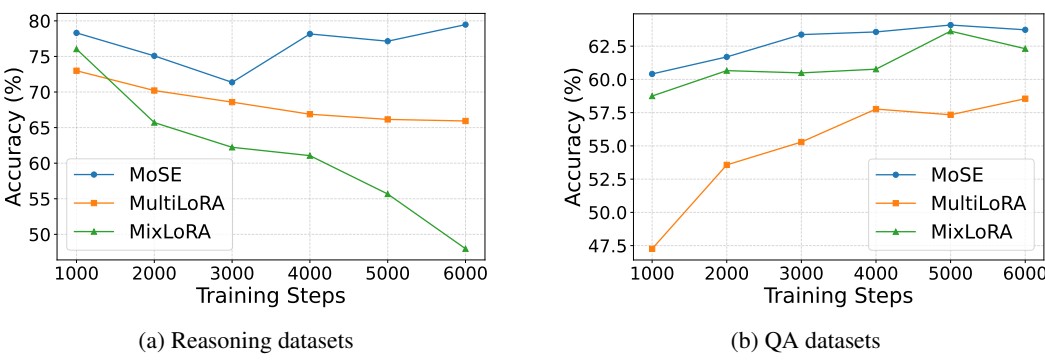

(a) Reasoning datasets                               (b) QA datasets

Figure 8: Accuracy trajectories during continual fine-tuning on QA tasks, starting from a model first supervised fine-tuned on commonsense reasoning datasets. The backbone is Qwen3-4B.

At the same time, we present the performance evolution of different tasks during sequential learning, as shown in Figure 9. Each cell reports the accuracy of a specific task after training on successive ones. Compared to MixLoRA and MultiLoRA, *MoSE* shows stronger resistance to forgetting and more stable performance preservation across tasks. In particular, even after the final task, *MoSE* maintains high accuracy, underscoring its effectiveness in retaining knowledge in multi-task continual learning scenarios.

## E.2    METRICS AND ANALYSIS

Moreover, we adopt two continual learning metrics: the *backward transfer*(Lopez-Paz & Ranzato, 2017) and *forgetting score*. Let $R_{i,j}$ denote the test accuracy on task $j$ immediately after completing task $i$, and let $T$ be the total number of tasks. The overall forgetting measure is defined in equation 13:

$$F = \frac{1}{T-2} \sum_{j=1}^{T-2} \left( \max_{k=j+1,...,T-1} R_{k,j} - R_{T,j} \right). \tag{13}$$

This metric quantifies how much the model's performance has degraded by comparing its best achievable accuracy under interference with its final performance. To complement this, we report the *backward transfer (BWT)*, which assesses whether learning new tasks helps or harms knowledge

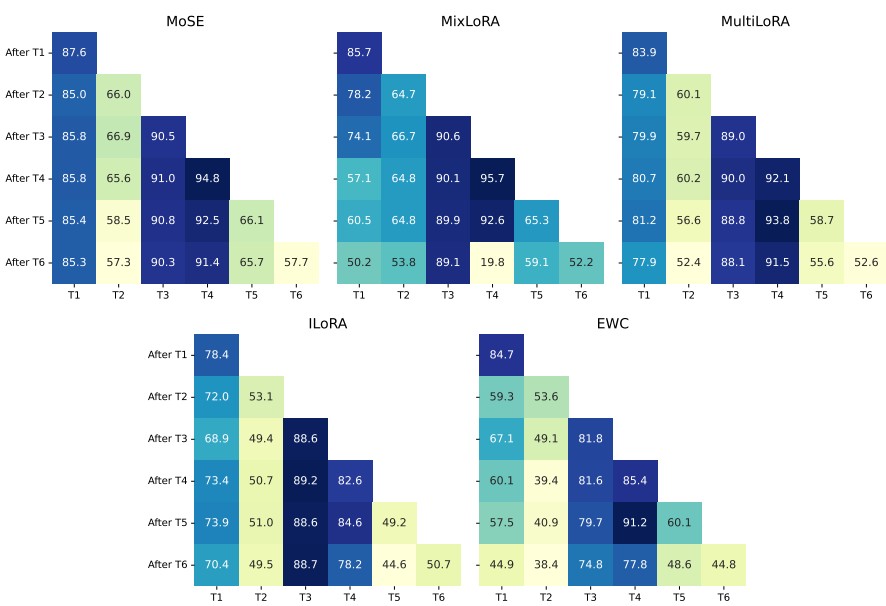

Figure 9: Continual learning performance across four task sequences: Reasoning (T1), QAs (T2), RACE (T3), SciTail (T4),FOMC(T5),MedMCQA(T6). The backbone is Qwen3-8B.

from previous tasks:

$$\text{BWT} = \frac{1}{T-1} \sum_{j=1}^{T-1} \left( R_{T,j} - R_{j,j} \right). \tag{14}$$

Together, these metrics provide a comprehensive view of a model's ability to preserve and transfer knowledge under sequential training.

Furthermore, following the standard continual learning protocol of (Lopez-Paz & Ranzato, 2017), we report the forgetting score and backward transfer for each task after conducting sequential training on Qwen3-8B. The results are presented in Table 15 and Table 16. As shown, *MoSE* achieves consistently low forgetting and near-neutral backward transfer across tasks, indicating strong stability throughout the sequence.

Table 15: Forgetting score on Qwen3-8B across sequential tasks (lower is better).

| Method | Reason | QAs | RACE | SciTail | Avg. |
|---|---|---|---|---|---|
| MoSE | 0.5 | 9.6 | 0.7 | 1.1 | **3.0** |
| MixLoRA | 28.0 | 12.9 | 1.0 | 72.8 | 28.7 |
| MultiLoRA | 3.3 | 7.8 | 1.9 | 2.3 | 3.8 |
| ILoRA | 3.5 | 1.5 | 0.5 | 6.4 | 3.0 |
| EWC | 22.2 | 10.7 | 6.8 | 13.4 | 13.28 |

Table 16: Backward transfer (BWT) performance on Qwen3-8B (higher is better).

| Method | Reason | QAs | RACE | SciTail | FOMC | Avg. |
|---|---|---|---|---|---|---|
| MoSE | -2.3 | -8.7 | -0.2 | -3.4 | -0.4 | **-3.0** |
| MixLoRA | -35.5 | -10.9 | -1.5 | -75.9 | -6.2 | -26.0 |
| MultiLoRA | -6.0 | -7.7 | -0.9 | -0.6 | -3.1 | -3.7 |
| ILoRA | -8.0 | -3.6 | 0.1 | -4.4 | -4.6 | -4.1 |
| EWC | -39.8 | -15.2 | -7.0 | -7.6 | -11.5 | -16.2 |

## F   LIMITATION

Despite strong empirical results, *MoSE* has several limitations. First, due to the device limitation, our experiments are limited to models up to 8B parameters, leaving scalability to larger models (e.g., 32B) unverified. Second, unlike standard LoRA, *MoSE*'s MoE-based design introduces additional parameters that cannot be merged at inference, increasing storage and latency. Finally, while effective on classification and reasoning tasks, its applicability to other domains such as code generation, long-form text, and multimodal learning remains unexplored.

## G   THE USE OF LARGE LANGUAGE MODELS

In preparing this work, we made limited use of large language models (LLMs) as an assistive tool. Specifically, LLMs were employed to:

- **Text refinement:** polishing grammar, enhancing readability, and improving academic style to ensure clarity and conciseness, without altering the substantive content or conclusions.
- **LaTeX support:** assisted in generating and formatting LaTeX code—covering equations, tables, and figure environments—to ensure consistency and improve typesetting quality.
- **Minor editing support:** providing suggestions for rephrasing section transitions, figure captions, and table descriptions.

LLMs were **not** used for any core contributions of this work, including research ideation, experimental design, algorithm development, or data analysis. All scientific content, methodology, and results originate solely from the authors.

