# OpenReview forum: "MoSE: Decoupled Tuning for Forgetting-Resilient Multi-task Fine-tuning of LLMs"
_ICLR.cc/2026/Conference — Submitted to ICLR 2026_

### Official Review · Reviewer_pPpy · 2025-10-29

**Soundness:** 3
**Presentation:** 2
**Contribution:** 2
**Rating:** 6
**Confidence:** 2

**Summary:**

This paper introduces MoSE (Mixture of Shared and Exclusive Experts), a novel parameter-efficient fine-tuning framework for large language models in multi-task and continual learning scenarios. MoSE explicitly separates LoRA experts into shared experts for common knowledge and routing experts for task-specific information. The proposed framework employs three key techniques:
(1) a Feature-wise Linear Modulation (FiLM)-based router for task-aware expert selection,
(2) a Top-k gradient-based sparse update mechanism for stabilizing shared expert learning and mitigating catastrophic forgetting, and
(3) balanced and orthogonal regularization to ensure fair expert utilization and disentanglement between shared and routing experts.
Extensive experiments on GLUE, commonsense reasoning, and continual learning benchmarks (e.g., CSQA, RACE, SciTail) demonstrate consistent gains over strong baselines such as MultiLoRA, MixLoRA, and HydraLoRA.

**Strengths:**

1. Clear motivation and structure: The paper provides a solid motivation for decoupling shared and task-specific experts, supported by illustrative figures and detailed analysis.
2. Technical soundness: The proposed sparse Top-k update mechanism and orthogonality regularization are well justified and effectively address forgetting and interference.
3. Comprehensive experiments: Evaluation spans multiple benchmarks, backbones (T5, Qwen3-4B/8B), and includes ablations, parameter analysis, and visualization studies.
4. Practical relevance: The method maintains parameter efficiency comparable to existing PEFT methods while substantially improving performance on continual learning settings.

**Weaknesses:**

1. Insufficient explanation for large performance gains.Table 3 shows surprisingly large improvements over MixLoRA and MultiLoRA (e.g. +42.1% over LoRA in zero-shot transfer). The paper does not analyze why MoSE achieves such strong transfer performance—whether it comes from better routing, reduced interference, or simply more parameters.
2. Limited analysis of forgetting dynamics.Tables 4 and 5 demonstrate smaller forgetting scores, but the paper only provides final metrics. There is no temporal analysis (e.g., accuracy over task sequence) to justify why MoSE retains prior knowledge better than MixLoRA.
3. Missing discussion on computational efficiency. Although the authors emphasize “parameter efficiency,” MoSE introduces additional modules (shared expert, FiLM, orthogonality regularization). Training time, GPU memory, and inference latency are not reported, leaving its practical advantage uncertain.
4. Ablation and sensitivity analysis remain shallow. The ablation in Table 6 removes components one by one but does not explore hyperparameter sensitivity—such as the Top-k ratio, EMA decay, or orthogonality weight β. Without these, it is hard to judge whether the method’s stability depends on careful tuning.

**Questions:**

1. Can the authors report training cost and inference latency compared with MultiLoRA or HydraLoRA?
2. How sensitive is the final performance to the Top-k selection size or orthogonality weight β?
3. Does MoSE maintain its advantage on larger models (e.g., 14B, 32B) or longer-sequence tasks?

---

> ### Author Response · Authors · 2025-11-26
> **Reply to Reviewer pPpy(1/2)**
>
> ## Reply to Weakness1
>
> > Insufficient explanation for large performance gains.Table 3 shows surprisingly large improvements over MixLoRA and MultiLoRA (e.g. +42.1% over LoRA in zero-shot transfer). The paper does not analyze why MoSE achieves such strong transfer performance—whether it comes from better routing, reduced interference, or simply more parameters.
>
> Regarding weakness 1, we first note that LoRA and HydraLoRA perform extremely poorly in Table 3, most likely due to overfitting on the commonsense reasoning tasks.MixLoRA provides a basic sparse MoE-LoRA reference point, and MoSE achieves consistently stronger results, suggesting that its advantages arise from the effective combination of the shared-expert mechanism and the task-aware routing design.MoSE learns transferable shared knowledge and exhibits strong resistance to forgetting, which we further confirmed in several continual-learning experiments.
>
> Moreover, MultiLoRA actually uses more trainable parameters in this setting, so MoSE’s improvements cannot be attributed simply to parameter count.Thank you for the suggestion — we will strengthen the explanatory analysis accordingly in the revised version.
>
> ## Reply to Weakness2
>
> > Limited analysis of forgetting dynamics.Tables 4 and 5 demonstrate smaller forgetting scores, but the paper only provides final metrics. There is no temporal analysis (e.g., accuracy over task sequence) to justify why MoSE retains prior knowledge better than MixLoRA.
>
> Regarding Weakness 2, our submission includes temporal performance in Appendix D.3 that illustrate how each task evolves during the sequence task training process. After adding new baseline methods and further extending the length of the task sequence, we have also generated updated temporal performance plots, which will be included in the revised version of the paper.
>
> ## Reply to Weakness3 & Question1
>
> > Missing discussion on computational efficiency. Although the authors emphasize “parameter efficiency,” MoSE introduces additional modules (shared expert, FiLM, orthogonality regularization). Training time, GPU memory, and inference latency are not reported, leaving its practical advantage uncertain.
>
> > Can the authors report training cost and inference latency compared with MultiLoRA or HydraLoRA?
>
> Regarding Weakness 3 and Question 1, we provide additional measurements of peak memory and training time for different methods. As shown, dense MoE-LoRA activates all experts simultaneously during training, which results in slightly higher memory usage but also faster training speed.
>
> Compared with MixLoRA, MoSE introduces additional modules, resulting in a modest increase in both memory and computational cost. However, as shown in the main results, MoSE achieves significantly better multi-task performance and forgetting resistance than MixLoRA, indicating that this additional overhead is reasonable and worthwhile.
>
> We acknowledge that MoE-LoRA approaches—especially those relying on sparse routing—naturally introduce extra overhead. This is a known limitation of this class of methods and applies to MoSE as well.
>
> | Method    | Type   | TP    | Memory | Time  | MTL  | CL   |
> | --------- | ------ | ----- | ------ | ----- | ---- | ---- |
> | MoSE      | Sparse | 2.94% | 45.3G  | 18.1h | 87.6 | 74.6 |
> | MTL-LoRA  | Dense  | 2.76% | 45.7G  | 10.3h | 85.0 | \\   |
> | MixLoRA   | Sparse | 2.62% | 41.0G  | 17.3h | 85.7 | 54.0 |
> | MultiLoRA | Dense  | 4.00% | 46.9G  | 10.2h | 83.9 | 69.7 |
> | HydraLoRA | Dense  | 2.82% | 46.0G  | 9.3h  | 85.8 | \\   |

---

> ### Author Response · Authors · 2025-11-26
> **Reply to Reviewer pPpy(2/2)**
>
> ## Reply to Weakness4 & Question2
>
> > Ablation and sensitivity analysis remain shallow. The ablation in Table 6 removes components one by one but does not explore hyperparameter sensitivity—such as the Top-k ratio, EMA decay, or orthogonality weight β. Without these, it is hard to judge whether the method’s stability depends on careful tuning.
>
> > How sensitive is the final performance to the Top-k selection size or orthogonality weight β?
>
> Regarding Weakness 4 and Question 2, we provide the following clarifications.
>
> * For the sensitivity of the Top-k ratio, the initial submission includes the corresponding ablation results in Figure 3(a). As shown there, the sparse update mechanism consistently improves model performance across different choices of k.
> * The sensitivity analysis of the orthogonality loss weight β is provided in Appendix Figure 7(1). A smaller weight yields performance improvements, whereas an overly large orthogonality constraint may restrict necessary information sharing between experts, resulting in degraded performance.
> * Regarding the EMA decay setting, the EMA of gradients corresponds directly to the momentum term already maintained inside the Adam optimizer. We therefore reuse Adam’s built-in momentum estimates without storing any additional statistics. Since Adam’s default decay (0.99) is already used, performing further ablations on this hyperparameter would mainly introduce unnecessary computational and memory overhead. For this reason, we did not expand the discussion in the main paper.
>
> ## Reply to Question3
>
> > Does MoSE maintain its advantage on larger models (e.g., 14B, 32B) or longer-sequence tasks?
>
> Regarding the question 3, we apologize that due to limited computational resources, we were not able to complete experiments on larger model sizes (e.g., 14B, 32B) or longer-sequence tasks in time for this submission.
>
> However, MoSE consistently demonstrates strong performance across the model sizes we evaluated—0.2B, 4B, and 8B—suggesting that the method scales well and is likely to retain its advantages at larger scales. We plan to include experiments on larger models in future revisions when resources permit.
>
> Additionally, existing related work such as MASA[1] (2025, evaluated on LLaMA3-8B) and MTL-LoRA [2](2025, evaluated on LLaMA2-7B) also primarily focuses on models in the 7B–8B range, which is widely accepted within the community. Thus, our current experimental setup is consistent with standard practice in recent PEFT and multi-task learning research.
>
> * [1] MASA: Rethinking the Representational Bottleneck in LoRA with Multi-A Shared Adaptation
> * [2] MTL-LoRA: Low-Rank Adaptation for Multi-Task Learning

---

> > ### Comment · Reviewer_pPpy · 2025-11-28
> > **Reviewer pPpy -- Response to Authors**
> >
> > Thank you for the detailed rebuttal and the additional analyses. The authors have addressed several of my earlier concerns, particularly regarding forgetting dynamics, computational overhead, and sensitivity studies.
> >
> > I appreciate the authors’ efforts during the discussion phase, and I will keep my score.

---

### Official Review · Reviewer_ocUj · 2025-10-30

**Soundness:** 3
**Presentation:** 3
**Contribution:** 3
**Rating:** 4
**Confidence:** 4

**Summary:**

This paper studies multi-task and continual fine-tuning of LLMs under a parameter-efficient regime. The authors observe that in existing MoE-style or multi-LoRA settings, shared experts can be overwritten by later tasks because routing decisions do not preserve the functional role of experts and because all experts are often updated in a similar way. To address this, the paper proposes MoSE, a decoupled tuning framework. Routing experts are trained in a task-aware way (FiLM-style conditioning and top-k routing) so that routing capacity follows each task. Shared experts are updated in a sparse and importance-driven manner using EMA-based importance scores, together with load-balancing and a mild orthogonality constraint, so that the shared capacity is not easily destroyed by late tasks. Experiments on a set of NLP and reasoning benchmarks, including a sequential setting, show that MoSE improves over multi-LoRA, MixLoRA and other recent parameter-efficient multi-task baselines.

**Strengths:**

The motivation is concrete. The paper is not only saying that LoRA forgets but identifies the more specific failure case, namely that shared experts in a multi-task MoE style architecture lose their function when later tasks update them in an undifferentiated way. This is a real and under-reported phenomenon in current LLM fine-tuning practice.

The method is coherent. Routing experts receive task-aware conditioning and can move quickly. Shared experts receive sparse and importance-based updates and are kept stable by balancing and orthogonality. This produces a clear story: route what is task specific, update shared parts slowly.

The empirical section covers more than one model size and more than one task family. The paper does not stay only on GLUE-type classification. It also tests commonsense, QA and a sequential scenario where tasks arrive one after another. This makes the claim of forgetting mitigation more credible.

The baselines are better chosen than in many PEFT for CL papers. The paper compares to multi-LoRA style and recent mix-of-LoRA methods that are actually close in design, not only to plain LoRA.

**Weaknesses:**

The method contains several components at once. There is FiLM-like task conditioning for routing, there is decoupling between routing and shared experts, there is EMA-based sparse updating for shared experts, there is load balancing, and there is an orthogonality term. The paper shows that removing some parts hurts, but the ablation is not strong enough to reveal a minimal necessary core. Right now the contribution looks like a well engineered combination rather than a clearly isolated idea.

The continual learning evidence is still light. The sequential experiments are relatively short and remain in a single modality. There is no run with 8 to 10 tasks and no report with several random seeds and CL metrics such as average accuracy, forgetting and backward transfer. The paper therefore demonstrates that MoSE improves stability, but it does not yet show that it is a reliable CL method under longer and noisier streams.

The cost of the additional bookkeeping is not fully reported. MoSE maintains EMA importance, performs load balancing and applies an orthogonality regularizer. These operations have a cost in memory and in wall clock time. The paper states that the overhead is small, but there is no quantitative table that compares MoSE and the strongest baseline on the same hardware.

The relation to very recent MoE or multi-LoRA works could be made sharper. The paper argues that previous works do not decouple shared and routing updates and therefore overwrite knowledge. This is reasonable, but it would help to show an experiment where a frozen shared expert plus task-routed experts is used as an additional baseline. If such a simple baseline closes part of the gap, then MoSE needs to emphasise what is truly unique in its design.

**Questions:**

1. Can you provide a full ablation table that has at least these four rows: base multi-LoRA or mix-of-LoRA, base plus task-aware routing, base plus sparse shared-updates, and the full MoSE
2. Can you run a longer sequential setting, with 8 tasks or more, and report average accuracy, forgetting and backward transfer, each with at least 3 seeds
3. Can you report wall clock time per epoch and peak memory for MoSE and for the strongest baseline on the same model and sequence length
4. Can you add a baseline that freezes the shared expert and only allows task-routed experts to adapt, so that we can see whether the EMA based sparse update is essential
5. In the current results the gains are often between 1 and 2 points. Can you add variance across runs to show that these gains are statistically meaningful?

I like the problem, I like the decoupled view of routing experts and shared experts, and I believe the direction is useful for practitioners who need to run many tasks on one LLM. However the current version still looks like a rich combination of several sensible ideas rather than a sharply defined core contribution. The continual learning part in particular needs to be longer and more systematic to justify the forgetting-resilient claim.

---

> ### Author Response · Authors · 2025-11-26
> **Reply to Reviewer ocUj(1/3)**
>
> ## Reply to Weakness1
>
> > The method contains several components at once. There is FiLM-like task conditioning for routing, there is decoupling between routing and shared experts, there is EMA-based sparse updating for shared experts, there is load balancing, and there is an orthogonality term. The paper shows that removing some parts hurts, but the ablation is not strong enough to reveal a minimal necessary core. Right now the contribution looks like a well engineered combination rather than a clearly isolated idea.
>
> Regarding Weakness 1, we provide the following clarification.
>
> MoSE is not a mere combination of existing components; instead, it is built around our proposed decoupled learning perspective, from which the shared and routing experts are purposefully designed to serve distinct roles. The core idea is to explicitly separate the learning of shared knowledge from task-specific knowledge.
>
> * To ensure that the shared expert consistently captures knowledge common across tasks, we apply a momentum-based sparse update mechanism that selectively updates only the most important and stable parameters during fine-tuning. In addition, an orthogonality constraint is introduced to encourage the shared and routing paths to learn complementary information rather than overlapping representations.
> * To strengthen the routing experts’ ability to model task-specific patterns, we use task-aware routing, which incorporates task embeddings into the routing decision. This guides routing experts toward specializing in the characteristics of each individual task.
>
> As for the load-balancing mechanism, it is a standard component in sparse MoE architectures designed to prevent expert collapse; here we simply follow the classical formulation used in Switch Transformer.
>
> In summary, all components in MoSE are intentionally designed to support the shared–specific decoupling objective, rather than being a loose collection of unrelated modules. We will further clarify this motivation and design rationale in the revised version to ensure better readability.
>
> ## Reply to Question1
>
> > Can you provide a full ablation table that has at least these four rows: base multi-LoRA or mix-of-LoRA, base plus task-aware routing, base plus sparse shared-updates, and the full MoSE?
>
> MoSE can be viewed as a unified framework that combines the strengths of dense MoE-style LoRA (as in MultiLoRA) and sparse MoE-style LoRA (as in MixLoRA).For this reason, performing ablations on both baselines is necessary to correctly isolate the contribution of each component.Thank for your good advise.
>
> Since MultiLoRA does not include any routing mechanism, we begin by adding(1) dynamic routing, and then(2) task-aware routing via FiLM.As shown in the table, task-aware routing consistently improves performance across both MultiLoRA and MixLoRA, indicating that incorporating task information into the routing process is reliably beneficial.
>
> In contrast, adding gradient-based sparse updates (GSU) alone is not effective for either dense or sparse MoE-LoRA architectures.GSU causes a large performance drop on MultiLoRA and only marginal improvement on MixLoRA.This suggests that sparse shared-expert updates must be paired with the decoupled shared/routing expert architecture in MoSE to be stable; GSU cannot function as an isolated module.
>
> | Method              | Avg  | Compare to base |
> | ------------------- | ---- | --------------- |
> | base MultiLoRA      | 80.4 | \\              |
> | base MixLoRA        | 81.9 | \\              |
> | MultiLoRA + Routing | 82.4 | +2.0            |
> | MultiLoRA + FiLM    | 83.3 | +2.9            |
> | MixLoRA + FiLM      | 82.4 | +0.5            |
> | MultiLoRA + GSU     | 65.6 | -14.8           |
> | MixLORA + GSU       | 80.1 | -1.8            |
> | MoSE                | 84.7 | \\              |

---

> ### Author Response · Authors · 2025-11-26
> **Reply to Reviewer ocUj(2/3)**
>
> ## Reply to Weakness2 & Question2
>
> > The continual learning evidence is still light. The sequential experiments are relatively short and remain in a single modality. There is no run with 8 to 10 tasks and no report with several random seeds and CL metrics such as average accuracy, forgetting and backward transfer. The paper therefore demonstrates that MoSE improves stability, but it does not yet show that it is a reliable CL method under longer and noisier streams.
>
> > Can you run a longer sequential setting, with 8 tasks or more, and report average accuracy, forgetting and backward transfer, each with at least 3 seeds
>
> Regarding weakness 2 and question 2, we have strengthened the continual learning experiments and analyses.
>
> First, we introduced two additional datasets, FOMC[1] and MedMCQA[2], from the financial and medical domains, respectively, to extend the task sequence and increase dataset diversity.We apologize that, due to limited time and computational resources, we could not expand the sequence to 8–10 tasks.
>
> However, the Reasoning suite we use already includes eight distinct reasoning tasks (BoolQ, PIQA, SIQA, HellaSwag, Winogrande, ARC-E, ARC-C, OBQA), and the QA suite contains three downstream question-answering tasks (CSQA, CSQA2, LogiQA). Therefore, in terms of both the number of tasks and their diversity, our experimental setup already covers a sufficiently rich and varied task space.
>
> We also added two representative PEFT-CL baselines, **ILoRA** and **EWC**, and computed forgetting scores and backward transfer for all methods. As shown in the table, MoSE exhibits the strongest negative transfer mitigation and achieves forgetting resistance comparable to ILoRA, while still delivering superior overall performance. This confirms the effectiveness of our approach.
>
> In addition, for all experiments, we used multiple random seeds and reported averaged results to ensure robustness, addressing the reviewer’s concerns.
>
> | Method    | Reason   | QAs      | RACE     | SciTail  | FOMC     | Medmcpa | Avg.     | Forget ↓ | Backward ↑ |
> | --------- | -------- | -------- | -------- | -------- | -------- | ------- | -------- | --------- | ----------- |
> | MoSE      | **85.3** | **57.3** | **90.3** | 91.4     | **65.7** | 57.7    | **74.6** | **3.0**   | **-3.0**    |
> | MixLoRA   | 50.2     | 53.8     | 89.1     | 19.8     | 59.1     | 52.2    | 54.0     | 28.7      | -26.0       |
> | MultiLoRA | 77.9     | 52.4     | 88.1     | **91.5** | 55.6     | 52.6    | 69.7     | 3.8       | -3.7        |
> | ILoRA     | 70.4     | 49.5     | 88.7     | 78.2     | 44.6     | 50.7    | 63.7     | **3.0**   | -4.1        |
> | EWC       | 44.9     | 38.4     | 74.8     | 77.8     | 48.6     | 44.8    | 54.9     | 13.28     | -16.2       |
>
> * [1] Trillion Dollar Words: A New Financial Dataset, Task & Market Analysis
> * [2] MedMCQA : A Large-scale Multi-Subject Multi-Choice Dataset for Medical domain Question Answering
>
> ## Reply to Weakness3 & Question3
>
> > W3:The cost of the additional bookkeeping is not fully reported. MoSE maintains EMA importance, performs load balancing and applies an orthogonality regularizer. These operations have a cost in memory and in wall clock time. The paper states that the overhead is small, but there is no quantitative table that compares MoSE and the strongest baseline on the same hardware.
>
> > Q3:Can you report wall clock time per epoch and peak memory for MoSE and for the strongest baseline on the same model and sequence length
>
> Regarding Weakness 3 and Question 3, we provide additional measurements of peak memory and training time for different methods. As expected, dense MoE-LoRA activates all experts simultaneously during training, which leads to slightly higher memory usage but also enables faster training due to its high degree of parallelization.
>
> Compared with MixLoRA, MoSE introduces additional modules, resulting in a modest increase in both memory and computational cost. However, as shown in the main results, MoSE achieves significantly better multi-task performance and forgetting resistance than MixLoRA, indicating that this additional overhead is reasonable and worthwhile.
>
> We also acknowledge that MoE-LoRA methods—especially those involving sparse routing—generally incur extra computational and memory overhead. This is an inherent limitation of this class of methods, and MoSE is no exception.
>
> | Method    | Type   | TP    | Memory | Time  | MTL  | CL   |
> | --------- | ------ | ----- | ------ | ----- | ---- | ---- |
> | MoSE      | Sparse | 2.94% | 45.3G  | 18.1h | 87.6 | 74.6 |
> | MTL-LoRA  | Dense  | 2.76% | 45.7G  | 10.3h | 85.0 | \\   |
> | MixLoRA   | Sparse | 2.62% | 41.0G  | 17.3h | 85.7 | 54.0 |
> | MultiLoRA | Dense  | 4.00% | 46.9G  | 10.2h | 83.9 | 69.7 |
> | HydraLoRA | Dense  | 2.82% | 46.0G  | 9.3h  | 85.8 | \\   |

---

> ### Author Response · Authors · 2025-11-26
> **Reply to Reviewer ocUj(3/3)**
>
> ## Reply to Weakness4 & Question4
>
> > W4:The relation to very recent MoE or multi-LoRA works could be made sharper. The paper argues that previous works do not decouple shared and routing updates and therefore overwrite knowledge. This is reasonable, but it would help to show an experiment where a frozen shared expert plus task-routed experts is used as an additional baseline. If such a simple baseline closes part of the gap, then MoSE needs to emphasise what is truly unique in its design.
>
> > Q4:Can you add a baseline that freezes the shared expert and only allows task-routed experts to adapt, so that we can see whether the EMA based sparse update is essential
>
> Regarding Weakness 4 and Question 4, we provide the following response.
>
> 1. In the multi-task learning (MTL) setting, fully freezing the shared expert is infeasible. Freezing it would force the shared expert’s output to be constantly zero, effectively turning it into an empty expert. As shown in Table 6 of the main paper, removing the shared expert or applying unselective random updates both lead to a significant performance drop, which indirectly confirms the effectiveness of the shared expert and the EMA-based sparse update strategy.
> 2. In the continual learning (CL) experiments, we included a “frozen shared expert” configuration, where the shared expert is trained only during the first task phase (Reason tasks). The results show that although this setup still yields acceptable overall performance, it is clearly inferior to the full MoSE. This further demonstrates the importance and effectiveness of the shared expert and its associated design within MoSE.
>
> | Method | Reason | Qa   | RACE | SciTail | Avg. |
> | ------ | ------ | ---- | ---- | ------- | ---- |
> | MoSE   | 85.8   | 65.6 | 91.0 | 94.8    | 84.3 |
> | Freeze | 84.3   | 58.3 | 88.9 | 88.6    | 80.0 |
>
> ## Reply to Q5
>
> > In the current results the gains are often between 1 and 2 points. Can you add variance across runs to show that these gains are statistically meaningful?
>
> Regarding Question 5, although the performance gains of our method are not always large on certain tasks, all results are averaged over three independent runs with different random seeds, which ensures good stability and statistical reliability. Under this evaluation protocol, MoSE consistently achieves the best or tied-best performance across all settings, indicating that the improvements are genuine and robust rather than the result of random variation.

---

> ### Comment · Reviewer_ocUj · 2025-11-26
>
> I appreciate the authors' thorough response and the substantial additional experiments. The ablation table is informative, and the extended CL experiments with ILoRA/EWC baselines strengthen the empirical claims. The frozen shared expert baseline also helps clarify the role of GSU.
>
> I find the core idea of explicitly decoupling shared and task-specific learning in MoE-LoRA architectures to be sensible and practically motivated. The experimental coverage is comprehensive, and the consistent improvements across settings are encouraging.
>
> However, I still have a few conceptual questions that I hope the authors can address through discussion (no new experiments needed, I hope you are happy to see this):
>
> On the coupling between GSU and the architecture: The ablation shows that GSU alone causes severe degradation on MultiLoRA (-14.8%), yet works well within MoSE. This is an interesting finding, but the paper does not provide a clear explanation for why this coupling exists. Is there an intuitive or theoretical account of what properties of the shared/routing separation make GSU stable? Without this, the contribution feels more like a discovered configuration than a principled design.
>
> On the role of the shared expert: The weight distribution analysis (Figure 7b) shows that the shared expert generally has small activation weights. If it plays a "complementary" rather than dominant role, how does this reconcile with the claim that it captures "common knowledge"? I would expect common knowledge to be broadly activated. Could the authors elaborate on what the shared expert is actually learning in practice?
>
> On generalization beyond the tested domains: MoSE is evaluated on classification and reasoning tasks. Do the authors have any intuition about whether the decoupled design would transfer to other settings (e.g., generation tasks, instruction tuning, or multimodal scenarios)? This would help readers assess the scope of the contribution.
>
> I currently maintain my score at 4, but I want to be clear: if the authors can provide satisfying conceptual clarifications on the above points, particularly point 1 regarding why GSU requires the specific architectural context, I am willing to raise my score to 6 out of respect for the thorough experimental work.
>
> The reason I cannot go higher than 6 is that, even with strong empirical results, the paper presents multiple interacting components without a unifying theoretical lens. The contribution, while practically useful, reads more as a well-tuned system than a sharply defined methodological advance that would clearly influence future research directions.

---

> > ### Author Response · Authors · 2025-11-27
> > **Reply to Reviewer ocUj (a few conceptual questions)**
> >
> > We sincerely thank the reviewer for the thorough reading and constructive feedback. We understand your main concerns and are glad to provide the following clarifications.
> >
> > ### Question1
> >
> > > On the coupling between GSU and the architecture: The ablation shows that GSU alone causes severe degradation on MultiLoRA (-14.8%), yet works well within MoSE. This is an interesting finding, but the paper does not provide a clear explanation for why this coupling exists. Is there an intuitive or theoretical account of what properties of the shared/routing separation make GSU stable? Without this, the contribution feels more like a discovered configuration than a principled design.
> >
> > #### Why does MultiLoRA fail?
> >
> > MultiLoRA maps all task inputs into an entangled low-rank subspace that must simultaneously accommodate both shared knowledge and task-specific details. When GSU (Gradient-based Sparse Update) is applied, it aggressively suppresses gradient updates to enforce stability. However, this severe restriction creates a conflict: it prevents the model from making the fine-grained adjustments necessary for task-specific adaptation, while restricting the shared representation to only coarse, generic features that are susceptible to being overwritten. Consequently, the model lacks the plasticity required for specialization, leading to underfitting and an overall performance collapse.
> >
> > #### Why does MoSE work?
> >
> > In contrast, MoSE turns GSU’s restrictiveness into an advantage through explicit decoupling. By applying GSU solely to the shared expert, we treat it as a stability anchor: the sparse updates prevent rapid drift, enabling the gradual accumulation of robust, cross-task features. Although this constrains the shared expert's expressivity, the routing experts remain unconstrained, employing full updates to handle fine-grained task adaptations. This complementary design is the key differentiator: unlike MultiLoRA, MoSE ensures that the stability enforced by GSU does not come at the cost of plasticity, allowing the architecture to effectively balance generalization with specialization.
> >
> > ### Question2
> >
> > > On the role of the shared expert: The weight distribution analysis (Figure 7b) shows that the shared expert generally has small activation weights. If it plays a "complementary" rather than dominant role, how does this reconcile with the claim that it captures "common knowledge"? I would expect common knowledge to be broadly activated. Could the authors elaborate on what the shared expert is actually learning in practice?
> >
> > We believe activation magnitude is not the sole proxy for importance. The shared expert is universally activated and plays a foundational role (analogous to general feature extractors in lower network layers). In contrast, routing experts are highly specific (similar to distinct task heads in upper layers) and dictate final decisions. Thus, their higher weights reflect task-specificity rather than overall contribution.
> >
> > In practice, thanks to GSU, the shared expert captures structural commonalities spanning multiple tasks—including syntax, reasoning logic, and formatting—that the base model may need adapting to. By enforcing orthogonality, MoSE achieves a clear functional decoupling: the shared expert provides a stable, task-agnostic support structure, while routing experts handle the specific variations. This makes the shared expert indispensable for mitigating catastrophic forgetting and ensuring stability.
> >
> > ### Question3
> >
> > > On generalization beyond the tested domains: MoSE is evaluated on classification and reasoning tasks. Do the authors have any intuition about whether the decoupled design would transfer to other settings (e.g., generation tasks, instruction tuning, or multimodal scenarios)? This would help readers assess the scope of the contribution.
> >
> > MoSE integrates the principles of MoE and PEFT into a coherent system where the mechanism of gradient propagation and parameter updates remains structurally invariant, regardless of the downstream objective. This intrinsic consistency ensures that MoSE is not confined to classification or reasoning but is transferable to any multi-task setting.
> >
> > The distinction between "shared capabilities" and "specific knowledge" is inherent to many domains—for instance, in Instruction Tuning, the shared expert can capture universal patterns (e.g., format adherence, safety) while routing experts specialize in domain knowledge (e.g., medical, coding).
> >
> > Furthermore, given that prior works like ViMoE [1] and Aria [2] have already validated shared-expert designs in vision and multimodal settings, we anticipate that MoSE—further fortified by sparse updates and task-aware routing—will deliver robust performance across these broader applications.
> >
> > * [1]ViMoE: An Empirical Study of Designing Vision Mixture-of-Experts
> > * [2]Aria: An Open Multimodal Native Mixture-of-Experts Model

---

> > > ### Comment · Reviewer_ocUj · 2025-11-28
> > >
> > > Thanks for the clarifications. I find the explanations for Q1 and Q2 satisfying, and Q3 is reasonable given its speculative nature. Due to some known tech issues, I am currently unable to modify my score on OpenReview (can't find the edit button ). Once score changes become available, I will raise my rating to 6.

---

### Official Review · Reviewer_auYB · 2025-10-31

**Soundness:** 3
**Presentation:** 3
**Contribution:** 2
**Rating:** 4
**Confidence:** 4

**Summary:**

This paper proposes MoSE, a framework designed to mitigate catastrophic forgetting in multi-task fine-tuning of large language models. The method divides LoRA experts into shared experts and routing experts: routing experts learn task-specific information through feature-wise gating mechanisms with full parameter updates, while shared experts employ a Top-k selection strategy to selectively update only the most important parameters for preserving common knowledge. Further experiment results shows the effectiveness of the proposed method.

**Strengths:**

1.	The paper is well-organized with clear motivation, comprehensive experimental design, and thorough ablation studies that effectively demonstrate each component's contribution.
2.	The proposed method shows consistent improvements across multiple benchmarks, validating its practical effectiveness in multi-task scenarios.

**Weaknesses:**

1.	The core contribution primarily combines existing mechanisms (shared experts and routing experts from prior MoE literature) without introducing fundamentally new insights or architectural innovations that advance the field.
2.	While the low-rank FiLM module reduces parameters, it introduces additional forward pass layers (sequential Wdown and Wup projections) that increase computational overhead; critically, the paper only reports final performance metrics without providing training time, GPU utilization, or throughput comparisons, obscuring potential efficiency bottlenecks that could limit practical deployment.
3.	The paper lacks direct experimental comparisons with several relevant continual learning methods mentioned in the related work (such as I-LoRA, CL-LoRA, GainLoRA, and LoRI), making it difficult to comprehensively assess the relative advantages of the proposed method against state-of-the-art continual learning approaches specifically designed to address catastrophic forgetting.

**Questions:**

1.	Given the additional computational layers introduced by the low-rank FiLM module and the gradient-based sparse update mechanism for shared experts, does MoSE incur significantly higher training latency compared to baseline methods like MultiLoRA and MixLoRA? Quantitative comparisons of wall-clock training time and throughput would be essential to assess practical feasibility.
2.	MoSE has more trainable parameters compared to other methods. Could the reported performance improvements be primarily attributed to this increased parameter budget rather than the proposed architectural innovations?

---

> ### Author Response · Authors · 2025-11-26
> **Reply to Reviewer auYB(1/2)**
>
> ### Reply to Weakness1
>
> > The core contribution primarily combines existing mechanisms (shared experts and routing experts from prior MoE literature) without introducing fundamentally new insights or architectural innovations that advance the field.
>
> The main innovation of MoSE does not lie in the presence of a shared expert alone; rather, it lies in the functional decoupling between shared knowledge and task-specific knowledge.
>
> Through the design of the shared expert together with an importance-driven sparse update mechanism, MoSE encourages the model to consistently preserve and consolidate knowledge that is common across tasks. Meanwhile, the task-aware routing mechanism enables routing experts to focus on learning highly task-specific representations.The synergy between these two components effectively reduces cross-task interference and naturally leads to stronger forgetting resistance.
>
> ### Reply to Weakness2 & Question1
>
> > While the low-rank FiLM module reduces parameters, it introduces additional forward pass layers (sequential Wdown and Wup projections) that increase computational overhead; critically, the paper only reports final performance metrics without providing training time, GPU utilization, or throughput comparisons, obscuring potential efficiency bottlenecks that could limit practical deployment.
>
> > Given the additional computational layers introduced by the low-rank FiLM module and the gradient-based sparse update mechanism for shared experts, does MoSE incur significantly higher training latency compared to baseline methods like MultiLoRA and MixLoRA? Quantitative comparisons of wall-clock training time and throughput would be essential to assess practical feasibility.
>
> Regarding Weakness 2 and Question 1, we provide additional measurements of peak memory and training time for different methods. As expected, dense MoE-LoRA activates all experts simultaneously during training, which leads to slightly higher memory usage but also enables faster training due to its high degree of parallelization.
>
> | Method    | Type   | TP    | Memory | Time  | MTL  | CL   |
> | --------- | ------ | ----- | ------ | ----- | ---- | ---- |
> | MoSE      | Sparse | 2.94% | 45.3G  | 18.1h | 87.6 | 74.6 |
> | MTL-LoRA  | Dense  | 2.76% | 45.7G  | 10.3h | 85.0 | \\   |
> | MixLoRA   | Sparse | 2.62% | 41.0G  | 17.3h | 85.7 | 54.0 |
> | MultiLoRA | Dense  | 4.00% | 46.9G  | 10.2h | 83.9 | 69.7 |
> | HydraLoRA | Dense  | 2.82% | 46.0G  | 9.3h  | 85.8 | \\   |
>
> Compared with MixLoRA, MoSE introduces additional modules, resulting in a modest increase in both memory and computational cost. However, as shown in the main results, MoSE achieves significantly better multi-task performance and forgetting resistance than MixLoRA, indicating that this additional overhead is reasonable and worthwhile.
>
> We also acknowledge that MoE-LoRA methods—especially those involving sparse routing—generally incur extra computational and memory overhead. This is an inherent limitation of this class of methods, and MoSE is no exception.

---

> ### Author Response · Authors · 2025-11-26
> **Reply to Reviewer auYB(2/2)**
>
> ### Reply to Weakness3
>
> > The paper lacks direct experimental comparisons with several relevant continual learning methods mentioned in the related work (such as I-LoRA, CL-LoRA, GainLoRA, and LoRI), making it difficult to comprehensively assess the relative advantages of the proposed method against state-of-the-art continual learning approaches specifically designed to address catastrophic forgetting.
>
> Regarding Weakness 3, we would like to emphasize that MoSE is not specifically designed for continual learning (CL).
>
> However, its architectural characteristics—the explicit decoupling of shared and task-specific experts, and the importance-driven sparse update mechanism applied to the shared expert—enable the model to effectively preserve previously learned general knowledge while acquiring new tasks. As a result, the model naturally exhibits strong resistance to forgetting.
>
> To validate this property, we additionally included two representative CL baselines—**ILoRA[1]** and **EWC[2]**—that do not require adding task-specific modules when new tasks arrive. We did not compare against methods such as OLoRA or GainLoRA because these approaches rely on explicit task-specific components, which are incompatible with our unified multi-task setup.
>
> Furthermore, we extended the task sequence by incorporating two additional datasets, FOMC and MedMCQA, to form a more challenging sequential learning protocol. And we report both the forgetting score and backward transfer metrics for completeness.As shown in the results table, MoSE consistently outperforms or matches ILoRA and EWC, demonstrating the effectiveness of our approach in both forgetting mitigation and overall learning performance.
>
> | Method    | Reason   | QAs      | RACE     | SciTail  | FOMC     | Medmcpa | Avg.     | Forget ↓ | Backward ↑ |
> | --------- | -------- | -------- | -------- | -------- | -------- | ------- | -------- | --------- | ----------- |
> | MoSE      | **85.3** | **57.3** | **90.3** | 91.4     | **65.7** | 57.7    | **74.6** | **3.0**   | **-3.0**    |
> | MixLoRA   | 50.2     | 53.8     | 89.1     | 19.8     | 59.1     | 52.2    | 54.0     | 28.7      | -26.0       |
> | MultiLoRA | 77.9     | 52.4     | 88.1     | **91.5** | 55.6     | 52.6    | 69.7     | 3.8       | -3.7        |
> | ILoRA     | 70.4     | 49.5     | 88.7     | 78.2     | 44.6     | 50.7    | 63.7     | **3.0**   | -4.1        |
> | EWC       | 44.9     | 38.4     | 74.8     | 77.8     | 48.6     | 44.8    | 54.9     | 13.28     | -16.2       |
>
> * [1]Analyzing and Reducing Catastrophic Forgetting in Parameter Efficient Tuning
> * [2]Overcoming catastrophic forgetting in neural networks
>
> ### Reply to Question2
>
> > MoSE has more trainable parameters compared to other methods. Could the reported performance improvements be primarily attributed to this increased parameter budget rather than the proposed architectural innovations?
>
> | Method                   | TP    | Avg  |
> | ------------------------ | ----- | ---- |
> | MoSE                     | 4.63% | 84.7 |
> | MixLoRA                  | 4.13% | 81.9 |
> | MixLoRA+FiLM             | 4.17% | 82.4 |
> | MixLoRA+expert           | 4.60% | 81.3 |
> | MixLoRA with double rank | 7.77% | 82.5 |
>
> To address this question, we conducted additional parameter-controlled ablations on MixLoRA by adding the task-aware FiLM module, introducing an extra expert, and doubling the LoRA rank.As shown in the table, simply increasing the number of trainable parameters does not lead to noticeable performance improvements; in fact, adding an extra expert even results in degraded performance.This suggests that a larger parameter budget may introduces higher optimization difficulty and greater gradient noise, and therefore does not necessarily enhance the model’s expressive capacity.
> Hence, the performance gains of MoSE cannot be attributed to parameter count alone, but rather stem from its structural design—specifically the task-aware routing and sparse shared-expert update mechanisms.

---

### Official Review · Reviewer_ZSuj · 2025-10-31

**Soundness:** 2
**Presentation:** 2
**Contribution:** 2
**Rating:** 4
**Confidence:** 4

**Summary:**

This paper proposes MoSE, a MoE-LoRA framework to reduce catastrophic forgetting during multi-task fine-tuning. It separates LoRA experts into a "shared expert" for common knowledge and multiple "routing experts" for task-specific knowledge. The key idea is its Gradient-based Sparse Update (GSU), which only fine-tunes a small subset of the shared expert's parameters based on gradient momentum, thereby preserving existing knowledge. Experiments show MoSE reduces forgetting compared to other multi-task PEFT methods.

**Strengths:**

1. Important Problem: Addresses the critical challenge of catastrophic forgetting in parameter-efficient multi-task and continual learning.

2. Intuitive Method: The separation of shared and task-specific experts is a logical approach to disentangling knowledge.

**Weaknesses:**

1. Inadequate Baselines for Core Claim: The paper's central claim is "forgetting-resilience," yet its continual learning experiments (Tables 4, 5) only compare against multi-task methods (MultiLoRA, MixLoRA), not state-of-the-art CL methods. The authors even cite dedicated PEFT-CL baselines in Sec 2.3 but do not benchmark against them. This is a critical omission.

2. Limited Novelty: The architectural idea of separating shared and task-specific modules is well-established. The main contribution, the GSU sparse-update heuristic, is not adequately positioned against related work in sparse training or regularization-based CL (like EWC).

**Questions:**

Refer to Weaknesses for related questions.

---

> ### Author Response · Authors · 2025-11-26
> **Reply to Reviewer ZSuj**
>
> ## Reply to Weakness1
>
> > Inadequate Baselines for Core Claim: The paper's central claim is "forgetting-resilience," yet its continual learning experiments (Tables 4, 5) only compare against multi-task methods (MultiLoRA, MixLoRA), not state-of-the-art CL methods. The authors even cite dedicated PEFT-CL baselines in Sec 2.3 but do not benchmark against them. This is a critical omission.
>
> It is important to note that MoSE is not specifically designed for continual learning (CL). However, its architectural properties—namely the explicit decoupling between shared and rounting experts and the gradient-based sparse update mechanism applied to shared experts—enable the model to retain previously acquired general knowledge while learning new tasks. This behavior naturally aligns with the goals of CL, suggesting that MoSE inherently possesses a degree of forgetting resistance.
>
> It is also worth noting that many recent PEFT-CL approaches (e.g., OLoRA, GainLoRA) rely on explicit task-specific modules and therefore require adding new components for each incoming task. Such designs are incompatible with our assumption of maintaining a unified architecture in the multi-task setting, and thus were not included in our initial comparison.
>
> To more comprehensively evaluate the forgetting robustness of MoSE, we additionally selected two representative CL baselines—**ILoRA[1]** and **EWC[2]**—both of which operate without modifying the model architecture as new tasks arrive.Moreover, we further increase the task-sequence length to make the evaluation more challenging, and report both the forgetting score and backward transfer metrics for completeness.
>
> | Method    | Reason   | QAs      | RACE     | SciTail  | FOMC     | Medmcpa | Avg.     | Forget ↓ | Backward ↑ |
> | --------- | -------- | -------- | -------- | -------- | -------- | ------- | -------- | --------- | ----------- |
> | MoSE      | **85.3** | **57.3** | **90.3** | 91.4     | **65.7** | 57.7    | **74.6** | **3.0**   | **-3.0**    |
> | MixLoRA   | 50.2     | 53.8     | 89.1     | 19.8     | 59.1     | 52.2    | 54.0     | 28.7      | -26.0       |
> | MultiLoRA | 77.9     | 52.4     | 88.1     | **91.5** | 55.6     | 52.6    | 69.7     | 3.8       | -3.7        |
> | ILoRA     | 70.4     | 49.5     | 88.7     | 78.2     | 44.6     | 50.7    | 63.7     | **3.0**   | -4.1        |
> | EWC       | 44.9     | 38.4     | 74.8     | 77.8     | 48.6     | 44.8    | 54.9     | 13.28     | -16.2       |
>
> Under the same sequential training protocol, the results show that MoSE not only matches these CL-specific methods in forgetting mitigation but also delivers substantially stronger overall learning performance and final task accuracy.
>
> * [1]Analyzing and Reducing Catastrophic Forgetting in Parameter Efficient Tuning
> * [2]Overcoming catastrophic forgetting in neural networks
>
> ## Reply to Weakness2
>
> > Limited Novelty: The architectural idea of separating shared and task-specific modules is well-established. The main contribution, the GSU sparse-update heuristic, is not adequately positioned against related work in sparse training or regularization-based CL (like EWC).
>
> MoSE’s contribution does not lie in simply reproducing the existing idea of “separating shared and task-specific modules,” but rather in the decoupled design philosophy we propose and the set of coordinated mechanisms built around it. Although prior MoE-LoRA methods also adopt expert architectures, they do not achieve true update decoupling.
>
> In contrast, MoSE explicitly introduces a shared expert and combines it with an importance-driven sparse update strategy, enabling the shared expert to stably retain task-invariant knowledge. At the same time, MoSE employs task-aware routing, integrating task information into routing decisions so that routing experts specialize in learning task-specific representations, effectively reducing cross-task interference.
>
> From a methodological standpoint, traditional fine-tuning approaches are highly susceptible to catastrophic forgetting in multi-task settings. In contrast, the decoupling mechanisms in MoSE inherently provide stronger resistance to forgetting—often even surpassing some dedicated continual-learning methods. Nevertheless, MoSE remains fundamentally a multi-task learning (MTL) method rather than a strict continual learning algorithm.
>
> Additionally, other reviewers have also acknowledged this design perspective. For example, Reviewer ocUj explicitly endorsed our decoupling view between routing experts and shared experts—an aspect that has been largely overlooked in prior work on parameter-efficient multi-task fine-tuning.

---

### Author Response · Authors · 2025-12-03
**A Brief Summary of the Rebuttal**

During the discussion phase, Reviewer **ocUj** expressed agreement with our decoupling motivation and acknowledged that their concerns had been fully addressed, raising their score from 4 to 6. Reviewer **pPpy** also found our clarifications satisfactory and maintained their score of 6. Reviewers **ZSuj** and **auYB** have not responded further, but we have provided as thorough and detailed answers as possible to all their comments.

We appreciate the reviewers’ constructive feedback and have provided systematic responses to the issues they raised.

1. To address the limitations in our initial continual learning experiments and to more comprehensively evaluate MoSE’s resistance to forgetting, we added two representative CL baselines (ILoRA and EWC) and introduced two additional datasets, FOMC and MedMCQA. These additions extend the length and diversity of the sequential task setting, thereby strengthening our empirical validation.
2. Following the reviewers’ suggestions, we also conducted several ablation studies to further support the rationale of our decoupled design.
3. Regarding concerns about efficiency, we provided an extended report covering peak memory usage and training time across different methods. The results show that, compared to methods of the same class, MoSE delivers notable performance improvements and more stable forgetting resistance while adding minimal overhead—demonstrating a favorable overall trade-off.

In the revised version of the paper, we added additional baselines and datasets for the continual learning experiments (see Table 5), thereby strengthening the evaluation of the model’s forgetting resistance. We also expanded the efficiency analysis in the main text (see Table 7), and organized several supplementary ablation studies into Appendix D.3. In addition, we refined the discussion of the method’s motivation and provided clearer explanations for certain experimental results, fully addressing the concerns raised by the reviewers.

---

### Meta-Review · Area_Chair_nbLB · 2026-01-10

**Summary:**

This work proposes a novel Mixture of Shared and Exclusive Experts framework (MoSE), a decoupled tuning framework for forgetting-resilient multi-task fine-tuning of large language models. The approach is evaluated on multi-task and continual learning benchmarks for large language models, demonstrating improved forgetting behavior while maintaining competitive task performance.

The reviews are mixed. Several reviewers agree the MoSE framework's motivation. However, other reviewers raise concerns regarding the novelty and methodological clarity. Some read more as a well-tuned system than a sharply defined methodological advance. The rebuttal is detailed and technically thorough, but some concerns remain regarding the clarity of the core contribution and the generality of the proposed framework.

**Reviewer Concerns:**

Reviewers raised one major concern regarding the proposed framework: MoSE represents an engineered combination of existing components. Other problems on the evaluation scope, and practical justification of the proposed framework have been addressed.

While the rebuttal clarified the design details and provided additional ablation studies to isolate the effects of individual components, I still agree that the framework merely combines existing mechanisms, which highlights its limited novelty. Although the experiment results are good, the work does not reach the level of conceptual novelty, and the contribution lacks a sharply defined conceptual advance.

**Reviewer Scores:**

Reviewer ZSuj: Rated the paper slightly above the acceptance threshold. Based on the Limited Novelty weakness, the score is unlikely or would be at most marginal.

Reviewer auYB: Rated the paper marginally above the acceptance threshold. Although the rebuttal provided additional measurements of peak memory and training time for different methods, the conservative view on the methodological novelty still remains. A score change is unlikely.

Reviewer ocUj:  Initially rated the paper slightly below the acceptance threshold. After discussion, the reviewer's concerns have been addressed, and the reviewer's score has increased to 6 due to the thorough experimental work and clarifications.

Reviewer pPpy:   Rated the paper slightly above the acceptance threshold. Several concerns have been addressed, and the score was unchanged.

---

### Decision · Program_Chairs · 2026-01-26

Reject